# Taking the neural sampling code very seriously: A data-driven approach for evaluating generative models of the visual system

**Suhas Shrinivasan**[1, †],   **Konstantin-Klemens Lurz**[1],   **Kelli Restivo**[2],
**George H. Denfield**[3],   **Andreas S. Tolias**[2, 5],   **Edgar Y. Walker**[4,*],   **Fabian H. Sinz**[1, 2*]

[1] Institute for Computer Science and Campus Institute for Data Science,
University of Göttingen, Göttingen, Germany
[2] Center for Neuroscience and Artificial Intelligence, Department of Neuroscience,
Baylor College of Medicine, Houston, USA
[3] Department of Psychiatry, Columbia University, New York City, USA
[4] Department of Physiology and Biophysics, and Computational Neuroscience Center,
University of Washington, Seattle, USA
[5] Department of Electrical and Computer Engineering, Rice University, Houston, USA
[†]Correspondence: `suhas.shrinivasan@uni-goettingen.de`, [*] Equal contribution

## Abstract

Prevailing theories of perception hypothesize that the brain implements perception via Bayesian inference in a generative model of the world. One prominent theory, the Neural Sampling Code (NSC), posits that neuronal responses to a stimulus represent samples from the posterior distribution over latent world state variables that cause the stimulus. Although theoretically elegant, NSC does not specify the exact form of the generative model or prescribe how to link the theory to recorded neuronal activity. Previous works assume simple generative models and test their qualitative agreement with neurophysiological data. Currently, there is no precise alignment of the normative theory with neuronal recordings, especially in response to natural stimuli, and a quantitative, experimental evaluation of models under NSC has been lacking. Here, we propose a novel formalization of NSC, that (a) allows us to directly fit NSC generative models to recorded neuronal activity in response to natural images, (b) formulate richer and more flexible generative models, and (c) employ standard metrics to quantitatively evaluate different generative models under NSC. Furthermore, we derive a stimulus-conditioned predictive model of neuronal responses from the trained generative model using our formalization that we compare to neural system identification models. We demonstrate our approach by fitting and comparing classical- and flexible deep learning-based generative models on population recordings from the macaque primary visual cortex (V1) to natural images, and show that the flexible models outperform classical models in both their generative- and predictive-model performance. Overall, our work is an important step towards a quantitative evaluation of NSC. It provides a framework that lets us *learn* the generative model directly from neuronal population recordings, paving the way for an experimentally-informed understanding of probabilistic computational principles underlying perception and behavior.

## 1   Introduction

Our environment is riddled with sensory stimuli that are noisy, ambiguous, and often incomplete, necessitating organisms to handle uncertainty in their sensory observations. Bayesian models of

37th Conference on Neural Information Processing Systems (NeurIPS 2023).

perception and behavior have thus grown in prominence, successfully accounting for an extensive array of tasks across perception [1, 2], cognition [3], sensory-motor learning [4] and decision making [5–8]. These models posit that the brain maintains a statistical generative model of the world, where sensory observations $\mathbf{x}$ are generated from unknown, world-state variable $\mathbf{z}$. Upon encountering a stimulus $\mathbf{x}$, perception in the brain is conceptualized as *probabilistically inferring* the world-state variable $\mathbf{z}$ that caused $\mathbf{x}$. In other words, to perceive $\mathbf{x}$, the brain would invert the generative model to compute the posterior distribution over $\mathbf{z}$: $p(\mathbf{z}|\mathbf{x})$. One can express the posterior via Bayes' rule as: $p(\mathbf{z}|\mathbf{x}) \propto p(\mathbf{x}|\mathbf{z}) p(\mathbf{z})$, where $p(\mathbf{z})$ is the prior distribution of $\mathbf{z}$ and $p(\mathbf{x}|\mathbf{z})$ is the conditional distribution characterizing how well a given $\mathbf{z}$ describes $\mathbf{x}$. While this has been an influential framework, the neuronal underpinnings of probabilistic inference remain challenging to conceptualize and test experimentally. To this end, the Neural Sampling Code (NSC) [9–16] is a prominent theory that offers a unique link between neuronal responses and probabilistic inference. Specifically, NSC posits that neuronal responses, $\mathbf{r}$, to a given stimulus, $\mathbf{x}$, can be thought of as samples drawn from the posterior distribution: $\mathbf{r} \sim p(\mathbf{z}|\mathbf{x})$ (Figure 1).

**Background and related work**   Prevailing literature on NSC uses simple and restrictive generative models and performs qualitative comparisons of model predictions with neurophysiological data to test the theory. Notably, existing NSC works use simple prior- and conditional distributions with pre-specified parameters. For example, a popular choice for the conditional distribution of images (stimuli) has been Gaussian with a likelihood function that linearly combines pre-specified filters. Hoyer and Hyvärinen [9] learn these filters via independent component analysis on natural images, whereas Haefner, Berkes, and Fiser [12] use oriented Gabor filters instead. Similarly, a popular choice for the prior is the exponential distribution with a pre-specified rate parameter [9]. These choices are inspired by (a) what is already known about sensory neurons, especially in the primary visual cortex (V1), and (b) the fact that it renders posterior computation mathematically simpler. In the examples above, the choice of filters reflects well-known findings that the receptive fields of V1 neurons resemble (Gabor-like) orientation filters [17–20], and the exponential prior is motivated by the principle of sparse coding [9, 20]. Importantly, these parameters and distributions — and thereby, the generative models — are not informed or learned *explicitly* from neurophysiological data. Rather, these works typically sample from the posterior of the assumed generative model in response to strongly parameterized stimuli (e.g., noisy oriented gratings). The models — and thereby the theory — are then evaluated based on how well the samples *qualitatively* capture specific neurophysiological phenomena such as the mean-variance relationship [9, 21, 22], task-induced noise correlation structures [12], and contextual modulation in V1 neurons [16].

In contrast, recent advances in deep learning-based neural system identification models have set new standards in providing expressive models that can faithfully predict neural population responses to natural stimuli [23–39], and offer experimentally verifiable insights at the single-neuron level [31, 40–42]. Additionally, advances in generative modeling, especially of images, have clearly demonstrated the effectiveness of deep, highly nonlinear, generative models such as auto-regressive models [43, 44], variational autoencoders [45–47], normalizing flows [48–51], and diffusion models [52–54]. Given the complexity of high-dimensional natural stimuli and real-world tasks, it is paramount that NSC be considered under a generative model that can match such complexity.

**Our objective and contributions**   Here we ask: what exactly is the brain's generative model? More specifically, can we identify the brain's generative model from NSC population responses to natural stimuli? Although simple generative models and qualitative evaluations in the NSC literature have offered us great insight into the potential generative models of the brain and engendered support for the theory, there remains a conspicuous gap in the quantitative evaluation of NSC, particularly in response to natural stimuli. In this work, we bridge this gap by proposing a formalization of NSC that ❶ allows us to directly fit NSC generative models to recorded neuronal activity in response to natural images, ❷ formulate richer and more flexible generative models, and ❸ employ standard metrics such as log-likelihood and single trial correlation to quantitatively evaluate different generative models under NSC. As opposed to specifying a generative model that ought to be maintained by the brain, our framework allows us to *learn* the generative model directly from neurophysiological data. Learning expressive generative models in a data-driven fashion additionally lets us take advantage of population recordings of large and ever-increasing scale in the field [55–57]. Furthermore, our formalization ❹ lets us derive a stimulus-conditioned predictive model of neuronal responses from the trained generative model, which can be directly compared to state-of-the-art system identification

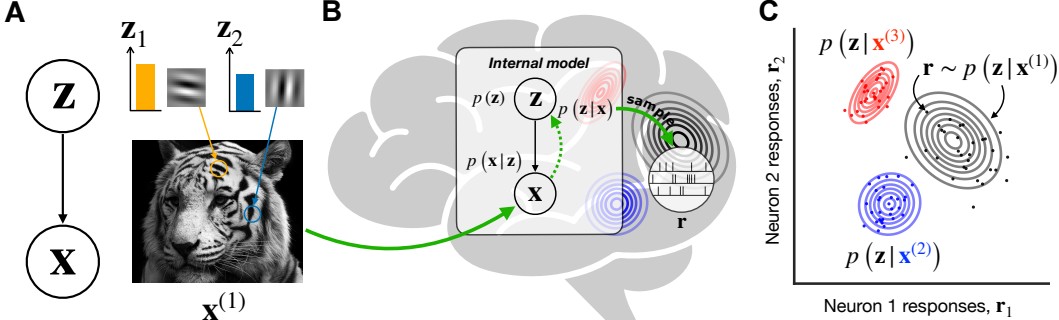

Figure 1: Conceptualizing NSC. **A.** Latent variable model of the world (stimulus): $\mathbf{z}$ is the world state variable (intensity of oriented Gabor filter here; figure inspired from Orbán et al. [13]) and $\mathbf{x}$ is the observed sensory stimulus (e.g., an image of a tiger). **B.** Responses $\mathbf{r}$ under NSC: As the brain encounters a stimulus $\mathbf{x}$, it inverts its generative model, combining the likelihood $p(\mathbf{x}|\mathbf{z})$ and prior $p(\mathbf{z})$, to obtain the posterior $p(\mathbf{z}|\mathbf{x})$, and $\mathbf{r}$ are samples from the posterior. **C.** Neural response distribution under NSC: Each point corresponds to a single response pair of two NSC neurons under three distinct stimuli depicted by distinct colors. The distribution of neurons matches the distribution over the corresponding latent variables $\mathbf{z}$.

models. The predictive model has the ability to provide experimentally-verifiable, neuron-specific predictions from the normative theory.

We demonstrate our approach by fitting classical generative models from NSC literature and flexible deep learning-based generative models on macaque primary visual cortex (V1) population responses to natural images. We show that the flexible models outperform classical models in both their generative- and predictive-model performance. Overall, this work presents an important step towards a quantitative evaluation of NSC, paving the way for a data-driven approach in *learning* the generative model of the brain.

## 2 Fitting the Neural Sampling Code

### 2.1 Theory

**An explicit formalization of NSC** We begin by formalizing NSC as a latent variable probabilistic model $\mathbf{z} \longrightarrow \mathbf{x} \longrightarrow \mathbf{r}$, where $\mathbf{z}$ represents the world state variable that underlies the observable stimulus $\mathbf{x}$ (Figure 1A). Subsequently, the stimulus $\mathbf{x}$ gives rise to the neuronal responses $\mathbf{r}$ via the posterior $p(\mathbf{z}|\mathbf{x})$ (Figure 1B). NSC posits that the neuronal responses $\mathbf{r}$ elicited by stimulus $\mathbf{x}$ can be interpreted as stochastic samples from the posterior distribution $p(\mathbf{z}|\mathbf{x})$ (Figure 1B, C). However, the exact relation between $\mathbf{z}$ and $\mathbf{r}$ is often left unspecified. For instance, it is not clear what aspect of the neuronal response (e.g., firing rate, presence or absence of spikes, or membrane potential) should be treated as a sample. In fact, most previous works do not make a distinction between $\mathbf{r}$ and $\mathbf{z}$, and simply equate an aspect of the neuronal response such as firing rate with the latent sample. Here, we make this assumption explicit and treat the neural response $\mathbf{r}$ as a random variable that matches the latent random variable $\mathbf{z}$ in stimulus-conditioned distributions:

$$\mathbf{z}_{\text{sample}} \sim p(\mathbf{z}|\mathbf{x}) \tag{1}$$

$$\mathbf{r} = \mathbf{z}_{\text{sample}}, \tag{2}$$

Equation 2 is a slight abuse of notation to state the equivalence in the stimulus-conditioned distributions of $\mathbf{r}$ and $\mathbf{z}$, more formally stated as:

$$p(\mathbf{r}|\mathbf{x}) \stackrel{\mathrm{d}}{=} p(\mathbf{z}|\mathbf{x}), \tag{3}$$

where $\stackrel{\mathrm{d}}{=}$ denotes equality in distribution or density function. By marginalizing the stimulus, we find that the marginal distribution of $\mathbf{r}$ must also match that of $\mathbf{z}$:

$$p(\mathbf{r}) = \int p(\mathbf{r}|\mathbf{x})\,p(\mathbf{x})\,\mathrm{d}\mathbf{x} \stackrel{\mathrm{d}}{=} \int p(\mathbf{z}|\mathbf{x})\,p(\mathbf{x})\,\mathrm{d}\mathbf{x} = p(\mathbf{z}). \tag{4}$$

Explicitly formalizing NSC with distinct $\mathbf{r}$ and $\mathbf{z}$ has two advantages. Firstly, the resulting formulation provides the crucial link between the generative model $\mathbf{z} \to \mathbf{x}$ and observed responses $\mathbf{r}$, serving as the basis for learning the generative model from the responses. This also provides the basis for a neuron-specific model comparison between different NSC models as well as the possibility to make predictions for specific neurons that can be experimentally tested. Secondly, the explicit link highlights the possibility to explore more flexible mappings between $\mathbf{z}$ and $\mathbf{r}$. For instance, one could assume that the latent variable $\mathbf{z}$ is encoded in the membrane potential, but what we observe are spike counts, i.e. $\mathbf{r} = f(\mathbf{z})$ for some stochastic mapping $f$. This relation can, in turn, become part of the model, which can then be fitted to real data and compared to alternative versions of the model. In this work, we choose to learn the generative models from the data under the simplest mapping of $\mathbf{r} \equiv \mathbf{z}$ (Section 3.2). Please see Section 4 for a discussion on alternative mappings between $\mathbf{r}$ and $\mathbf{z}$.

In NSC, we note that the latent variables are what underlie the stimulus (such as the intensity of an oriented filter in 1A) and are not necessarily any task-relevant experimenter-defined variables (such as orientation in an orientation-discrimination task). This is in contrast to the alternative theory of probabilistic population codes [58, 59], where typically, the latent variable is explicitly defined to be the task-relevant experimenter-defined variables.

**Learning the generative model under NSC**  One way to quantitatively test an NSC generative model $p(\mathbf{z}, \mathbf{x})$ is testing how well the response distribution $p(\mathbf{r}|\mathbf{x})$ approximates the posterior $p(\mathbf{z}|\mathbf{x})$ of the generative model, i.e., testing equation 3. However in reality $p(\mathbf{z}, \mathbf{x})$, and consequently $p(\mathbf{z}|\mathbf{x})$, is unknown to us. However, our formalization allows us to learn the generative model $p(\mathbf{z}, \mathbf{x})$ via learning the joint distribution $p(\mathbf{r}, \mathbf{x})$. The equivalence between $p(\mathbf{z}, \mathbf{x})$ and $p(\mathbf{r}, \mathbf{x})$ follows from equations 2, 3 and 4:

$$p(\mathbf{z}, \mathbf{x}) = p(\mathbf{z}|\mathbf{x}) p(\mathbf{x}) \stackrel{\mathrm{d}}{=} p(\mathbf{r}|\mathbf{x}) p(\mathbf{x}) = p(\mathbf{r}, \mathbf{x}) \tag{5}$$

$$= p(\mathbf{x}|\mathbf{z}) p(\mathbf{z}) \stackrel{\mathrm{d}}{=} p(\mathbf{x}|\mathbf{r}) p(\mathbf{r}) = p(\mathbf{r}, \mathbf{x}) \tag{6}$$

The joint distribution can then simply be fitted to recorded stimulus-response pairs $\{\mathbf{x}^{(i)}, \mathbf{r}^{(i)}\}_{i=1}^N$ by maximizing the likelihood:

$$\theta^* = \arg\max_\theta \sum_{i=1}^N \log \underbrace{p\left(\mathbf{x}^{(i)}, \mathbf{r}^{(i)}; \theta\right)}_{\text{Joint}} = \arg\max_{\theta_L, \theta_P} \sum_{i=1}^N \log \underbrace{p\left(\mathbf{x}^{(i)}|\mathbf{r}^{(i)}; \theta_L\right)}_{\text{Likelihood}} + \log \underbrace{p\left(\mathbf{r}^{(i)}; \theta_P\right)}_{\text{Prior}} \tag{7}$$

where $\theta$ are the parameters of the generative model, that we split into $\theta_L$ and $\theta_P$ for parameters relevant to the likelihood and prior, respectively. Provided that $\theta_L$ and $\theta_P$ do not overlap, the generative models can be learned by learning the likelihood and prior separately.

**Evaluating NSC on data**  Fitting generative models under NSC on recorded data allows us to compare the generative models quantitatively by evaluating their performances as log-likelihood on a held-out test set. Furthermore, once we have learned the generative model $p(\mathbf{r}, \mathbf{x}; \theta^*)$, we can invert it to arrive at the posterior $p(\mathbf{r}|\mathbf{x}; \theta^*)$. This provides a neuronal encoding model $p(\mathbf{r}|\mathbf{x})$ under specific assumptions of the NSC model, allowing us to predict neural responses to arbitrary new stimuli. The performance of this predictive model can serve as yet another metric for quantitative model comparison under NSC. Additionally, the posterior allows us to compare the generative models to the normative-theory-free system identification models. It is important to note that our quantification does not make any assumption about the kind of stimuli $\mathbf{x}$. Existing works on NSC use parametric stimuli from classical neuroscience experiments and perform a qualitative comparison between model- and real-neuronal responses to the same stimuli. Our formulation, on the other hand, allows us to compare different NSC models on *natural images*, the type of stimuli the visual system has evolved to process.

## 2.2 Models

Following previous work in NSC [9–16], here we focus on vision and develop generative models under NSC for natural image stimuli $\mathbf{x}$ and spike counts $\mathbf{r}$ recorded from the visual cortex (Figure 2A). Developing generative models entails developing models for the prior $p(\mathbf{r})$ and the likelihood $p(\mathbf{x}|\mathbf{r})$ (Figure 2B). Additionally, we also fit an approximate posterior $q(\mathbf{r}|\mathbf{x})$. We summarize our fitting methodology in an algorithm towards the end of the section (Algorithm 1).

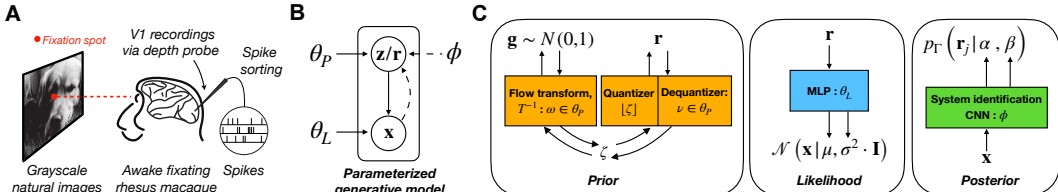

Figure 2: **A.** Overview of the experimental setup for neuronal recordings (see Section 3.2 for a summary of the data description and Cadena et al. [29] for the complete details). **B.** Parameterized generative model for NSC: $\theta_P$: parameter of the prior $p\left(\mathbf{r};\theta_P\right)$; $\theta_L$: parameter of the likelihood $p\left(\mathbf{x}|\mathbf{z};\theta_L\right)$; $\phi$: parameters of the (approximate) posterior $q\left(\mathbf{r}|\mathbf{x};\phi\right)$. **C.** Flexible prior, likelihood, and posterior models. The prior follows our dequantization framework consisting of 3 components: (1) continuous prior $p(\boldsymbol{\zeta};\omega)$ (normalizing flow with Gaussian base distribution), (2) quantizer $P(\mathbf{r}|\boldsymbol{\zeta})$ (floor function), and (3) variational dequantizer $q\left(\boldsymbol{\zeta}|\mathbf{r};\nu\right)$ (a conditional normalizing flow, Appendix B). The likelihood is an isotropic Gaussian distribution over $\mathbf{x}$, where the parameters are functions (MLP) of $\mathbf{r}$. The posterior is a Gamma distribution over $\mathbf{r}$ whose parameters are functions of $\mathbf{x}$, modeled as a system identification-based convolutional neural-network model (Appendix C).

**Prior**   Spike counts $\mathbf{r}$ are discrete. Hence, we can neither directly fit standard literature models, such as an exponential [9] or a Laplace distribution [20] to the discrete variable $\mathbf{r}$, nor can we straightforwardly fit more common flexible density models such as normalizing flows [49, 60], as they are all continuous density models. A common approach to remedy this is to employ uniform dequantization, where the discrete quantity is converted into a continuous signal by adding uniform random noise [43, 61, 62]. We adopt the more general approach of variational dequantization where the noise distribution is learned instead of being fixed to be uniform [63–66]. In this method, prior distribution over discrete $\mathbf{r}$ is captured by positing a generative model involving a continuous latent variable $\boldsymbol{\zeta}$ linked to the discrete response $\mathbf{r}$ via a deterministic quantizer function: $P\left(\mathbf{r}\right) = \int P(\mathbf{r}|\boldsymbol{\zeta})p(\boldsymbol{\zeta})d\boldsymbol{\zeta}$, where $P(\mathbf{r}|\boldsymbol{\zeta})$ is the *quantizer* and $p(\boldsymbol{\zeta})$ is the *continuous prior*.

Since the integral is usually intractable, the whole model is fit by optimizing the evidence lower bound (ELBO):

$$\log P\left(\mathbf{r}\right) \geq \mathbb{E}_{\boldsymbol{\zeta}\sim q(\boldsymbol{\zeta}|\mathbf{r};\nu)}\left[\log \overbrace{p\left(\boldsymbol{\zeta};\omega\right)}^{\text{Continuous prior}}\right] + \mathbb{H}\left(\overbrace{q\left(\boldsymbol{\zeta}|\mathbf{r};\nu\right)}^{\text{Dequantizer}}\right) \qquad (8)$$

where $q\left(\boldsymbol{\zeta}|\mathbf{r};\nu\right)$ is the approximate posterior distribution with parameters $\nu$, $p\left(\boldsymbol{\zeta};\omega\right)$ is the continuous prior with parameters $\omega$, and $\mathbb{H}\left(q\left(\boldsymbol{\zeta}|\mathbf{r};\nu\right)\right) = -\mathbb{E}_{\boldsymbol{\zeta}\sim q(\boldsymbol{\zeta}|\mathbf{r};\nu)}\left[\log q\left(\boldsymbol{\zeta}|\mathbf{r};\nu\right)\right]$ is the conditional entropy of the dequantizing distribution. Note that Equation (8) only provides a lower-bound to $\log P\left(\mathbf{r}\right)$, and a tighter bound via importance-weighted sampling [66–68] (Appendix A).

In our work, we only consider factorized prior distributions, i.e., we treat neurons to be *a priori* independent $\log P\left(\mathbf{r}\right) = \sum_i \log P\left(\mathbf{r}_i\right)$. This choice is informed by both the nature of the V1 neural data that showed limited correlation across all stimuli and the simplicity of the fit it provides. The same independence assumption was applied for the continuous prior distribution over the dequantized responses $\boldsymbol{\zeta}$. Given this dequantizer framework, we explore three different NSC priors by varying the distribution over the continuous latent $p\left(\boldsymbol{\zeta};\nu\right)$:

1. Exponential (**Exp**), $\frac{1}{\lambda}\exp\frac{-\boldsymbol{\zeta}}{\lambda}H\left(\boldsymbol{\zeta}\right)$, as found in the original NSC model by Hoyer & Hyvärinen [9].

2. Half-normal (**HN**), $\frac{\sqrt{2}}{\sigma\sqrt{\pi}}\exp\left(-\frac{\boldsymbol{\zeta}^2}{2\sigma^2}\right)H\left(\boldsymbol{\zeta}\right)$, where $H\left(\boldsymbol{\zeta}\right)$ is the heavyside function.

3. Normalizing flow (**Flow**): $p(\boldsymbol{\zeta};\omega) = p_{\text{base}}(T^{-1}(\boldsymbol{\zeta};\omega))\cdot|\frac{\partial\,T^{-1}(\boldsymbol{\zeta};\omega)}{\partial\boldsymbol{\zeta}}|$, where we choose $p_{\text{base}}$ to be a standard normal, and $T^{-1}$ represents the following series of invertible mappings with learnable parameters $\omega$: [affine, $\tanh$, affine, $\tanh$, affine, $\tanh$, affine, softplus$^{-1}$], where softplus$^{-1}(y) = \log(e^y - 1)$, affine$(y) = ay + b$ with learnable parameters $a$ and $b$. softplus$^{-1}$ ensures that the support of $\boldsymbol{\zeta}$ is non-negative, since we are ultimately interested in modeling the distribution of (non-negative) spike counts (Figure 2C, "Prior" sub-panel).

For the dequantizer distribution, $q\left(\boldsymbol{\zeta}|\mathbf{r};\nu\right)$, we utilize a conditional normalizing flow-based flexible distribution, as in [66] (Appendix B).

**Likelihood**   We model the likelihood as an isotropic Gaussian distribution:

$$p\left(\mathbf{x}|\mathbf{r}^{(i)}\right) = \mathcal{N}\left(\mathbf{x}|\boldsymbol{\mu}^{(i)}, \boldsymbol{\sigma}^{2(i)}\cdot\mathbf{I}\right), \tag{9}$$

where the parameters mean, $\boldsymbol{\mu}^{(i)}\in\mathbb{R}^{|\mathbf{x}|}$ and variance, $\boldsymbol{\sigma}^{(i)}\in\mathbb{R}_{>0}^{|\mathbf{x}|}$ are functions of response, $\mathbf{r}^{(i)}$, and $|\mathbf{x}|$ is the number of dimensions of $\mathbf{x}$. We consider (1) a linear function where $\boldsymbol{\mu}=w_{\boldsymbol{\mu}}\mathbf{r}^{(i)}+b_{\boldsymbol{\mu}}$ and $\boldsymbol{\sigma}=\exp^{w_{\boldsymbol{\sigma}}\mathbf{r}^{(i)}+b_{\boldsymbol{\sigma}}}$ (**Lin**) and (2) a nonlinear function $\boldsymbol{\mu}=w_{\boldsymbol{\mu}}\text{MLP}(\mathbf{r}^{(i)})+b_{\boldsymbol{\mu}}$ and $\boldsymbol{\sigma}=\exp^{w_{\boldsymbol{\sigma}}\text{MLP}(\mathbf{r}^{(i)})+b_{\boldsymbol{\sigma}}}$, where $\text{MLP}(\cdot)$ is a neural network (**MLP**) (Figure 2C "Likelihood" sub-panel).

**Posterior**   In many cases, the posterior distribution for a desired generative model is not analytically tractable and must be approximated, commonly using variational inference or Markov Chain Monte Carlo sampling [69–71]. Here, since we learn the generative model $p\left(\mathbf{x},\mathbf{r};\theta^*\right)$, we can approximate the true posterior $p\left(\mathbf{r}|\mathbf{x};\theta^*\right)$ by fitting a model posterior $q\left(\mathbf{r}|\mathbf{x};\phi\right)$ to samples from $p\left(\mathbf{x},\mathbf{r};\theta^*\right)$ directly via maximum log-likelihood:

$$\phi^* = \arg\max_{\phi}\sum_{\mathbf{x}',\mathbf{r}'}\log q\left(\mathbf{r}'|\mathbf{x}';\phi\right), \tag{10}$$

where $\mathbf{x}',\mathbf{r}'\sim p\left(\mathbf{x},\mathbf{r};\theta^*\right)$, are samples from the trained generative model. We model the posterior distribution of responses conditioned on images as a factorized Gamma distribution, following state-of-the-art (SOTA) work in system identification [72]: $p\left(\mathbf{r}|\mathbf{x}^{(i)}\right) = \prod_{j=1}^{S}p_{\Gamma}\left(\mathbf{r}_j|\boldsymbol{\alpha}^{(i)},\boldsymbol{\beta}^{(i)}\right)$, where $\mathbf{x}^{(i)}$ is the $i$th image, $\mathbf{r}_j$ is the $j$th neuron out of $|\mathbf{r}|=S$ total neurons, and the parameters concentration, $\boldsymbol{\alpha}^{(i)}$ and rate, $\boldsymbol{\beta}^{(i)}$ are functions of the image, $\mathbf{x}^{(i)}$. Since these functions map an image to response distribution parameters, we model them using a convolutional neural network model (Figure 2C "Posterior" sub-panel), following SOTA system identification work [25, 27, 73] (Appendix C).

| Prior | Likelihood | Name |
|---|---|---|
| Exponential ($\lambda=1$) | Linear | Exp1-Lin |
| Exponential | Linear | Exp-Lin |
| Half-Normal | Linear | HN-Lin |
| Normalizing Flow | Linear | Flow-Lin |
| Exponential | MLP | Exp-MLP |
| Half-Normal | MLP | HN-MLP |
| Normalizing Flow | MLP | Flow-MLP |

Table 1: Generative models that we fit as being composed of priors and likelihoods.

---

**Algorithm 1** Learning NSC models from data

**Require:** $N$ pairs of stimuli and neuronal responses respectively, $\{\mathbf{x}^{(i)},\mathbf{r}^{(i)}\}_i^N$
**Learning generative model** $p\left(\mathbf{x},\mathbf{r};\theta_P,\theta_L\right)$
1: $\theta_P^* \leftarrow \arg\max_\theta\sum_{i=1}^N\log p\left(\mathbf{r}^{(i)};\theta_P\right)$
2: $\theta_L^* \leftarrow \arg\max_\theta\sum_{i=1}^N\log p\left(\mathbf{x}^{(i)}|\mathbf{r}^{(i)};\theta_L\right)$
**Learning approx posterior model** $q\left(\mathbf{r}|\mathbf{x};\phi\right)$
3: Sample $\{\mathbf{x}'^{(i)},\mathbf{r}'^{(i)}\}_i^S \sim p\left(\mathbf{x},\mathbf{r};\theta_P^*,\theta_L^*\right)$
4: $\phi^* \leftarrow \arg\max_\phi\sum_i^S\log q\left(\mathbf{r}'^{(i)}|\mathbf{x}'^{(i)};\phi\right)$

---

## 3   Experiments

### 3.1   Synthetic data

We simulated 10,000 pairs of images and neuronal responses from the following three classical NSC models: ❶ a Hoyer & Hyvärinen model (HNH) with an exponential prior [9], ❷ an Olshausen & Field (ONF) model where the prior is a Laplace distribution [20], and ❸ a full Gaussian model (Gauss) where the prior is an isotropic Gaussian with mean 0 and variance $\sigma_r^2$. All the three models share a common linear, isotropic Gaussian likelihood $p\left(\mathbf{x}|\mathbf{r}\right)=\mathcal{N}\left(\mathbf{x}|A\mathbf{r},\sigma^2\mathbf{I}\right)$, where $A$ is the factor loading matrix learned via standard independent component analysis model (ICA) with a complete basis on natural image patches [9, 74]. Additionally, we sampled image-response pairs from ❹ our flexible model with **Flow** prior (described in Section 2.2) and MLP-based likelihood (Section 2.2), where all parameters were randomly initialized. For any given generative model, we first sample neuronal responses from the prior via $\mathbf{r}^{(i)}\sim p\left(\mathbf{r}\right)$, and then sample corresponding images via

$\mathbf{x}^{(i)} \sim p\left(\mathbf{x}|\mathbf{r}^{(i)}\right)$, where $i \in 1, \ldots, 10,000$. We hold out a set of 1,000 pairs as the test set. We fitted all models on the datasets simulated from the classical as well as the flexible models via Equation (7) and computed joint log-likelihoods of the trained models on the held-out test set as $\log p\left(\mathbf{x}^{(i)}, \mathbf{r}^{(i)}\right) = \log p\left(\mathbf{x}^{(i)}|\mathbf{r}^{(i)}\right) + \log p\left(\mathbf{r}^{(i)}\right)$. For the classical models, maximum likelihood estimates of the parameters were obtained analytically (Appendix F). We trained the flexible model using gradient descent.

We find that (1) the flexible model fits responses and images simulated under other NSC models well, i.e., learns $p(\mathbf{r})$ and $p(\mathbf{x}|\mathbf{r})$ and closely approximates the log-likelihood of the true models. Importantly, it outperforms the fit of other NSC models with mismatched generative distributions, consistently being the best model after the ground-truth model (first 3 columns in Figure 3). Furthermore, the flexible model is capable of generating complex image and response distributions that could not be easily captured by the classical generative models (column 4 in Figure 3). This demonstrates that our framework allows for NSC model fitting and that the flexible model has the ability to flexibly capture the data distribution across a wide range of generative models. Critically, a flexible model could fit complex generative models that cannot be modeled well by other classical models.

Figure 3: $\log p(\mathbf{x}, \mathbf{r})$ of models on simulated data (trial averaged, in bits). Column denotes the model generating the samples (data) and rows the trained NSC model. Since the exponential prior in HNH has a non-negative support and does not match that of ONF and Gauss, the scores for HNH under ONF and Gauss data are unavailable.

## 3.2 Neurophysiological data

**Data description** Next, we demonstrate applying our approach to real neuronal data. We used 32-channel laminar NeuroNexus arrays (Figure 2**A**) to record population activity from the primary visual cortex (V1) of two awake male rhesus macaque monkeys (*Macaca mulatta*) [29] as they fixated on grayscale natural images sampled from the ImageNet dataset [75]. All the experiments concerned with the recordings adhered to the National Institutes of Health, United States guidelines, and received approval from the Institutional Animal Care and Use Committee. Each image was presented for 120 ms, and spike counts between 40 ms and 160 ms after the image onset were computed and used as the neuronal response $\mathbf{r}$. The image stimulus $\mathbf{x}$ used for modeling is $41 \times 41$ px. For more details on the experiments and data collection, refer to Cadena et al. [29]. We collected data across 12 recording sessions, each having approximately 16,000 image-response pairs and at least 16 well-isolated single units. We split the dataset into approximately 10,000 pairs for training, 3,000 pairs for validation, and 3,000 for testing (for exact details on all sessions, see Appendix D). We do not aggregate data across sessions and fit models separately for each session since the images can differ from session to session, and not all neurons have seen every image.

**Fitting the generative model** Given a dataset of images and responses, $\{\mathbf{x}, \mathbf{r}\}_{i=0}^{N}$, we fit the likelihood $p(\mathbf{x}|\mathbf{r}; \theta_L)$ on the image-response pairs and the prior $p(\mathbf{r}; \theta_P)$ on the responses, $\mathbf{r}$ as in Equation (7). We fit all of the generative models on each recording session as following procedure described in Table 1. Below, we describe our results for the session with the largest number of neurons (29 well-isolated single units) in detail and report summary results on all sessions.

For the prior models (Figure 4**A**), we report the test-set log-likelihood performance of all models (Exp, HN, Flow) relative to the Exp1-model as the baseline. We find that our flexible normalizing flow model (Flow) achieves the best performance, improving the score from the exponential distribution (Exp) by 0.095 bits per neuron per trial, amounting to 2.755 bits across 29 neurons per trial. For the likelihood models (Figure 4**B**), we find that using an MLP likelihood function, the model improved by 0.052 bits per pixel per trial, amounting to 87.19 bits of improvement across all $41 \times 41$ pixels per trial, relative to the model with linear likelihood function. For the joint distribution (Figure 4**C**), we find that the flexible model (Flow-MLP) achieves the highest log-likelihood score, offering an improvement of 1.8452e-3 bits per pixel per neuron per trial, amounting to 89.951 bits across all $41 \times 41$ pixels and 29 neurons per trial. We observed that in each of the cases, flexible models (Flow prior, MLP likelihood) offer much higher log-likelihood performance, with the same trend found across all sessions (Appendix D).

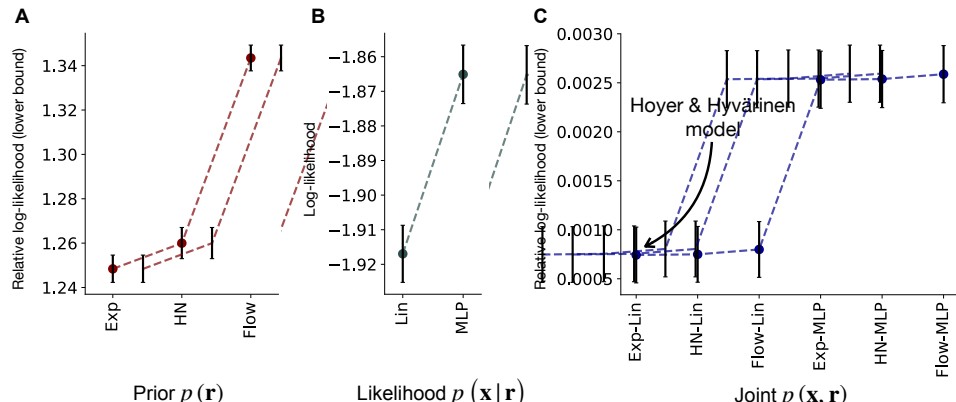

Figure 4: Log-likelihood scores (in bits) of generative models on population recordings (test set) as fit on recording session with the highest number of neurons ($n = 29$). Error-bars denote the standard error of mean across trials. **A**. Prior models, $p(\mathbf{r})$: log-likelihood (lower bound) relative to the baseline Exp1 prior model. The score is averaged across neurons and trials. Note that for the prior models on discrete spike counts, $\mathbf{r}$, we can only obtain a lower bound on $p(\mathbf{r})$. Here we show the importance-sampling bound (Equation (11)) with 1000 samples. **B**. Likelihood models $p(\mathbf{x}|\mathbf{r})$: absolute log-likelihood of likelihood functions, averaged across image pixels and trials. **C**. Joint models $p(\mathbf{x}, \mathbf{r})$: log-likelihoods relative to the baseline Exp1-Lin generative model. The score is averaged across pixels, neurons and trials.

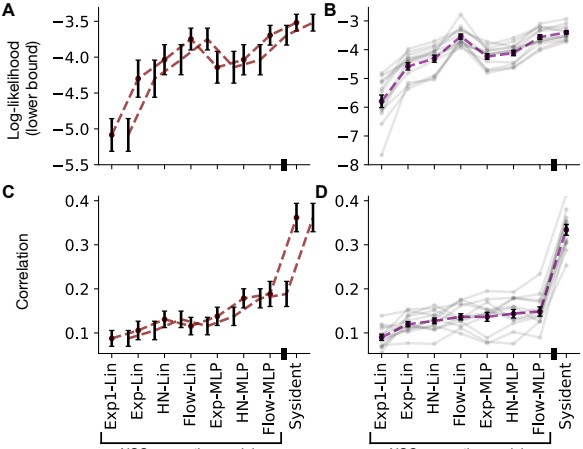

Figure 5: Posterior performance of NSC models, along with system identification model ("Sysident"). Error bars represent the standard error of mean over neuronal responses. All metrics are averaged over number of neurons and trials. **A**. The lower bound of log-likelihood in bits, for session with 29 neurons. We compute the lower-bound since we are evaluating the Gamma-posterior on (discrete) spike counts, and full-likelihood is intractable (Appendix G). **B**. Same as A but across all 12 sessions (purple: average, gray: single session). **C**. Single-trial correlation, for session with 29 neurons. **D**. Same as C across all 12 sessions

**Learning the posterior distribution** For each of the trained generative models, $p(\mathbf{x}, \mathbf{r}; \theta^*)$, we approximated the model's posterior distribution $p(\mathbf{r}|\mathbf{x}; \theta^*)$ using an approximate posterior $q(\mathbf{r}|\mathbf{x}; \phi)$ trained on samples drawn from the trained generative model (Equation (10), Algorithm 1). We evaluated the posterior distribution for each generative model by computing their mean log-likelihood on real neuronal responses conditioned on real images from the test set (Figure 5**A, B**). We also computed a single-trial correlation between the mean of the learned posterior distributions and neuronal responses (Figure 5**C, D**). Finally, we compared the posterior distribution to a deep system identification model.

We find that, in general, a more flexible trained generative model tends to yield a higher posterior predictive performance. Based on the log-likelihood evaluation, our flexible generative model (Flex-MLP) gained as much as 1.39 bits per neuron per trial compared to the baseline Exp1-Lin model and 0.61 bits per neuron per trial compared to Exp-Lin model (the Hoyer & Hyvärinen model [9]). In terms of single-trial correlation performance, our flexible generative model (Flex-MLP) achieved 10% higher correlation compared to the Exp1-Lin baseline and 8% higher correlation compared to Exp-Lin. When averaged across all sessions, we find that Flow-Lin performs best, almost on

par with the Flow-MLP, which achieves 0.019 bits per neuron per trial less. Furthermore, a system identification model trained on the dataset of real neuronal responses performed better than the best NSC generative model, gaining 0.17 bits per neuron per trial and 17% higher correlation per neuron per trial compared to Flex-MLP.

All model training was performed using backpropagation and gradient descent and we provide training, compute and infrastructure details in Appendix E.

## 4 Discussion

The main focus of this work was to develop a way to answer the question, "How well does NSC explain neurophysiological data *quantitatively*?". While NSC is a prominent normative theory for probablistic computation in the brain, and the literature has provided much qualitative insight, our work is the first to offer a quantitative paradigm for empirically testing it using brain responses to ecological, natural stimuli. Our framework additionally lets us formulate more flexible generative models — which can be better informed by the data — and employ standard metrics such as log-likelihood to quantitatively evaluate alternative generative models under NSC. Furthermore, inverting the learned generative model has allowed us to obtain the posterior distribution, which is equivalently a neuronal response predictive model. Importantly, this let us compare NSC models to models outside of NSC's theoretical framework, such as system identification models, allowing us to benchmark the predictive performance of NSC models. Our results demonstrated that the flexible generative models outperformed classical models in terms of both generative and predictive model performance, yet system identification models achieve superior response-predictive performance compared to even our best generative models. We now discuss some limitations of our current study and discuss a number of open questions and imminent future directions.

**Limitations I: Assumption of strict 1:1 neuron-latent mapping**: One limitation in our current study is that we only use a 1:1 identity mapping between the activity of neurons and latent variables in our formulation of NSC. Abiding by this restriction could limit the capacity of the NSC models, especially considering some existing work in NSC that have qualitatively explored more flexible mappings. For example, Orbán et al. [13] model membrane potential values (responses) as a nonlinear function of posterior samples. Furthermore, Savin and Denève [76] map responses of $N$ neurons to $D$ latent variables where $N > D$. Many more ways of how $\mathbf{r}$ and $\mathbf{z}$ relate are conceivable. However, our formulation with a separate $\mathbf{r}$ and $\mathbf{z}$ allows us to, in principle, incorporate different mappings and learn the corresponding generative models. Since the focus of this study was on the aspect of fitting NSC models to data, we chose the simplest (original) interpretation of NSC where $\mathbf{r} \equiv \mathbf{z}$.

**Limitations II: Definition of a "sample" as total spike counts**: We defined a "sample" as the total spike count of neuronal activity within a specific time window following the stimulus, which is not necessarily what literature works do. However, to our knowledge, there is no generally agreed upon or rigorous definition of a "sample" in NSC. While NSC was originally motivated with firing rate/spike counts over a 500ms window as the sample [9, 21], many alternative definitions such as membrane potential over 10ms [13] have been employed. It is unclear on what generally applicable metric — other than goodness of fit to data — such a definition could be evaluated. This in fact served to us as another motivating factor for striving towards a data-driven evaluation of sampling models that would allow one to compare such choices in an informed manner. In this work, we chose the total spike count as the working definition.

**Limitations III: Better generative models are needed:** Advances in deep learning architectures, latent variable models, and transfer learning have greatly enhanced the capabilities of generative models in machine learning. We believe the models we chose, although more expressive than classical models, are still limiting, especially considering that our likelihood $p(\mathbf{x}|\mathbf{r})$ uses linear or MLP decoding from neurons to images, with a simple Gaussian noise model. To capture the rich and complex nature of neuronal representations of natural images, we believe it is necessary to consider more sophisticated generative models, that even incorporate a natural image prior, that would eventually close the gap in predictive performance between system identification performance and NSC generative models. Furthermore, an important avenue of research is identifying biological mechanisms that underlie NSC (i.e. sampling from the posterior) [14, 77–82]. It is worth noting that our deep learning-based generative models are not meant to be mechanistic models of NSC neurons.

Rather, we believe that our approach lays the foundation for alternative biologically plausible models to be quantitatively evaluated and compared.

**Why do system identification models perform better than NSC generative models?** System identification models are directly trained discriminatively, i.e., $\min_\theta \mathbb{E}_{\mathbf{x}} \left[ D_{\text{KL}} \left( p_{\text{true}} \left( \mathbf{r}|\mathbf{x} \right) || p_{\text{model}} \left( \mathbf{r}|\mathbf{x}; \theta \right) \right) \right]$, to predict neuronal responses to natural images and deep-learning based ones are currently SOTA. There is still much room to build better generative models that would better explain the data (see Limitations III). However, for a given dataset of responses to stimuli from a *fixed stimulus distribution*, we do not expect the posterior of even the ideal generative model to surpass the performance of the ideal system identification model because the generative model training, i.e., $\min_\theta \left\{ \mathbb{E}_{\mathbf{x}} \left[ D_{\text{KL}} \left( p_{\text{true}} \left( \mathbf{r}|\mathbf{x} \right) || p_{\text{model}} \left( \mathbf{r}|\mathbf{x}; \theta \right) \right) \right] + D_{\text{KL}} \left( p_{\text{true}} \left( \mathbf{x} \right) || p_{\text{model}} \left( \mathbf{x}; \theta \right) \right) \right\}$, does not provide any advantage over system identification in response prediction unless some specific inductive biases are introduced in the generative model.

**Why bother fitting NSC models if they fail to quantitatively compete with system identification?** If we change the stimulus distribution $p \left( \mathbf{x} \right)$ to $p_{\text{new}} \left( \mathbf{x} \right)$ with markedly different stimulus statistics and let the sensory neurons adapt to $p_{\text{new}} \left( \mathbf{x} \right)$, we would expect the system identification model's performance to drop on $p_{\text{new}} \left( \mathbf{x} \right)$. The system identification model might have to be retrained on a new dataset of responses under $p_{\text{new}} \left( \mathbf{x} \right)$. This is the case where we would expect the NSC's learned generative model to be beneficial. Specifically, change in $p \left( \mathbf{x} \right)$ to $p_{\text{new}} \left( \mathbf{x} \right)$ may entirely derive from the change in prior $p \left( \mathbf{z} \right)$ to $p_{\text{new}} \left( \mathbf{z} \right)$, while $p \left( \mathbf{x}|\mathbf{z} \right)$ remains fixed. Hypothetically, this is since $p \left( \mathbf{x}|\mathbf{z} \right)$ represents the invariant "physical" process by which the latents (e.g., the identity of an animal) give rise to observations (e.g., the appearance of the animal). Consequently, if NSC accurately describes a neural population, i.e, $\mathbf{r} \sim p \left( \mathbf{z}|\mathbf{x} \right) \propto p \left( \mathbf{x}|\mathbf{z} \right) p \left( \mathbf{z} \right)$, the neuronal adaptation can be accounted for by simply learning $p_{\text{new}} \left( \mathbf{z} \right)$, i.e., $\mathbf{r}_{\text{new}} \sim p_{\text{new}} \left( \mathbf{z}|\mathbf{x} \right) \propto p \left( \mathbf{x}|\mathbf{z} \right) p_{\text{new}} \left( \mathbf{z} \right)$, keeping $p \left( \mathbf{x}|\mathbf{z} \right)$ fixed. We believe such out-of-context generalization is a theoretical strength of NSC, and is a consequence of its normative nature (responses being "samples" from the *posterior distribution*). Such normative hypotheses are neither present in the purely phenomenological system identification models and nor is it straightforward to equip them with normative assumptions.

The above insight thus helps us identify potential future experiments to test NSC models utilizing our framework since it lets us learn the generative model $p \left( \mathbf{x}|\mathbf{z} \right) p \left( \mathbf{z} \right)$ via $p \left( \mathbf{x}|\mathbf{r} \right) p \left( \mathbf{r} \right)$ (NSC assumption). Namely, one could perform experiments in which we let the neural population adapt to different sensory contexts with expected shifts in $p \left( \mathbf{z} \right)$. Using our NSC framework, we would expect to be able to predict how neuronal responses should change (as reflected in updated $\mathbf{r}_{\text{new}} \sim p_{\text{new}} \left( \mathbf{z}|\mathbf{x} \right)$) under new contexts.

**Why have previous works not fit NSC models to data?** We attribute the lack of such attempts to (1) limitations in data availability, (2) complexities involved in training flexible machine learning and inference algorithms on recorded data, and (3) the philosophical approach behind normative theories. Normative theories describe how a biological system *ought* to function in order to tackle fundamental tasks. They propose models with parameters that are optimized for those tasks, without relying on actual experimental data [83, 84]. Typically normative theories are evaluated using qualitative agreements between the proposed models and data. NSC is itself a normative theory. In contrast, phenomenological approaches such as system identification propose models whose parameters are directly learned from experimental data. Normative and phenomenological approaches have historically been developed independently of each other. Similar to Młynarski et al. [83], who interpolate between phenomenological and normative models via maximum entropy priors, our approach allows us to get the best of both worlds: state-of-the-art deep learning-based system identification models from phenomological approaches and the theoretical underpinnings of the normative NSC. System identification provides us with expressive models that faithfully model and predict the activity of thousands of neurons to rich natural stimuli. NSC, on the other hand, goes beyond what experimental data alone could offer by letting us hypothesize how neurons encode uncertainty about the stimulus, reflecting the posterior distribution over latent variables in a generative model of the world, thus allowing us make novel predictions such as about generalizability across stimulus contexts and design relevant experiments.

## Acknowledgements

We thank all the reviewers for their valuable and constructive feedback. We additionally thank Jakob Macke, Xaq Pitkow, Ralf Haefner, Gergő Orbán, members of Sinz-, Walker-, Tolias-lab for helpful and stimulating discussions. SS and FHS are supported by the German Research Foundation (DFG): SFB 1233, Robust Vision: Inference Principles and Neural Mechanisms, TP 06, project number: 276693517. KKL is supported by German Federal Ministry of Education and Research through the Tübingen AI Center (FKZ: 01IS18039A). KR and AST are supported by the National Eye Institute, National Institutes of Health (NIH), USA with award numbers R01 EY026927, and Core Grant for Vision Research with grant number T32-EY-002520-37. AST, KR and FHS are supported by the National Science Foundation Collaborative Research in Computational Neuroscience, USA with grant number IIS-2113173, Germany with FKZ: 01GQ2107. GHD is supported by The National Institute of Mental Health, NIH, USA with grant number T32MH015144. EYW is supported by the National Institute of Neurological Disorders and Stroke, NIH, USA with grant number 1U19NS107609-01.

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
