# Appendices

## A  Importance-weighted sampling estimate

Note that Equation (8) only provides a lower-bound to $\log P(\mathbf{r})$, and a tighter bound via importance-weighted sampling. Following Hoogeboom, Cohen, and Tomczak [66], Burda, Grosse, and Salakhutdinov [67], and Domke and Sheldon [68], we computed the importance-weighted sampling estimate as follows:

$$\log P(\mathbf{r}) \geq \log \left[ \frac{1}{K} \sum_{k=1}^{K} \frac{p(\boldsymbol{\zeta}_k; \omega)}{q(\boldsymbol{\zeta}_k|\mathbf{r}; \nu)} \right], \tag{11}$$

where $\boldsymbol{\zeta}_k \sim q(\boldsymbol{\zeta}|\mathbf{r}; \nu)$, $q(\boldsymbol{\zeta}|\mathbf{r}; \nu)$ is the approximate posterior distribution with parameters $\nu$; $p(\boldsymbol{\zeta}; \omega)$ is the continuous prior with parameters $\omega$, and $K$ is the number of samples.

## B  Model for the dequantizer distribution

For the dequantizer distribution, $q(\boldsymbol{\zeta}|\mathbf{r}; \nu)$, we utilize a conditional normalizing flow-based flexible distribution, as in [66].

The conditional normalizing flow utilizes a series of bijective affine transformations with ELU nonlinearity. It has a base distribution that follows a conditional isotropic Gaussian distribution. Mathematically, we can express $q(\boldsymbol{\zeta}_j|\mathbf{r}; \nu)$, where $\boldsymbol{\zeta}_j$ is the continuous latent variable behind neuronal response $\mathbf{r}_j$, as:

$$q(\boldsymbol{\zeta}_j|\mathbf{r}; \nu) = \mathcal{N}\left(\boldsymbol{\eta}_j = g_d(\boldsymbol{\zeta}_j; \xi)\right)|\boldsymbol{\mu}_j, \boldsymbol{\sigma}_j^2 \mathbf{I}) \cdot |\mathbf{J}_d| \tag{12}$$

In this equation, $\mathbf{J}_d = \frac{\partial \boldsymbol{\eta}_j}{\partial \boldsymbol{\zeta}_j}$ represents the Jacobian matrix of the transformation function $g_d$ with respect to $\boldsymbol{\zeta}_j$. $\boldsymbol{\mu}_j = w_{\boldsymbol{\mu}_j}\mathrm{MLP}(\mathbf{r}) + b_{\boldsymbol{\mu}_j}$ and $\boldsymbol{\sigma}_j^2 = \exp^{w_{\boldsymbol{\sigma}_j}\mathrm{MLP}(\mathbf{r})+b_{\boldsymbol{\sigma}_j}}$ and MLP (multi-layer perceptron) is a multi-layered neural network with nonlinearities.

Note that $\boldsymbol{\mu}_j$ and $\boldsymbol{\sigma}_j^2$, being dependent on $\mathbf{r}$, is what makes the dequantizing distribution a "conditional" normalizing flow, and we refer to $\boldsymbol{\mu}_j$ and $\boldsymbol{\sigma}_j^2$ as the "conditioning functions".

## C  System identification model architecture

This work uses a system idenfitication model for two purposes: (1) as the approximate posterior distribution's amortization function that maps images to neuronal responses (refer to paragraph on posterior models under Section 2.2) and (2) as a standalone model of neuronal responses conditioned on images that can be compared to the approximate posterior (Figure 5).

The system identification model used in this work is an artificial neural network similar to the ones proposed by Cadena et al. [25], Lurz et al. [27] and Baroni et al. [73]. It consists of a nonlinear "core" network and a linear "readout" network: the core is shared amongst all neurons and is used to fit the non-linear features of the image that are encoded by the neurons. It consists of 3 convolutional layers with 32 features channels, each being followed by a batch normalization and an ELU nonlinearity. The latter two convolutional layers are depth-separable. In the readout network, the relevant features in the core are selected for each neuron separately. This selection is done using a pyramid readout network (Sinz et al. [39]) which learns a spatial location (x,y) and extracts the features in the core at this location of the last layer as well as of two progressively down-sampled versions (average pooling, kernel size of 3) of the last layer's output. In Sinz et al. [39], the so obtained features at location (x,y) are then weighed by a neuron-specific feature vector and passed through a non-linearity to obtain the neuron specific firing rate of the Poisson distribution. Differently to this, we model the neural responses with a Gamma distribution which has two parameters: a rate and concentration parameter, which both have the constraint of being non-negative. We thus learn two locations and two weight

| Session ID | Neuron-count | Train-size | Validation-size | Test-size |
|:---:|:---:|:---:|:---:|:---:|
| A | 24 | 9826 | 2399 | 2820 |
| B | 20 | 9110 | 2319 | 2715 |
| C | 18 | 12020 | 2995 | 3915 |
| D | 17 | 5604 | 1341 | 2070 |
| E | 29 | 10779 | 2795 | 3690 |
| F | 18 | 14196 | 3548 | 4770 |
| G | 22 | 9732 | 2387 | 3000 |
| H | 17 | 10084 | 2546 | 3735 |
| I | 21 | 10529 | 2596 | 3255 |
| J | 21 | 9654 | 2390 | 3000 |
| K | 25 | 11814 | 2901 | 4050 |
| L | 20 | 6478 | 1651 | 2145 |

Table 2: Summary of the recorded data from all sessions. The data consists of image-response pairs of neurons from monkey V1. Images were cropped to 41x41 pixels and normalized to match the mean and standard deviation of the train and validation images.

vectors per neuron to account for the two parameters and pass each resulting value through a ELU + 1 non-linearity to ensure both outputs to be positive valued. Our model is similar to the Zero-Inflated Gamma model of Lurz et al. [72], differing only in that it does not model the zero inflated version of the Gamma distribution and in that it uses the pyramid readout mechanism instead of the Gaussian readout.

## D Details of data and models on all sessions

In our section on experiments on recorded neuronal response data (Section 3.2), we defer the exact details of all the recording sessions to the appendix. Please refer to Table 2 for a summary of details on all recording sessions. Additionally, in our section on experiments on recorded neuronal response data (Section 3.2), we only provide results of the fit of the generative models (prior, likelihood and joint, Figure 4) as fit on the session with the highest number of neurons (session ID "E" with 29 neurons (Table 2)). Please refer to Figure 6, Figure 7, Figure 8 for fits of the generative models on all sessions.

## E Model training and hyperparameter searches

We train all models using the PyTorch library [85], using the Adam optimizer [86]. Since for us, the generative model consists of three different models: the prior, likelihood and posterior (Section 2.2), we describe the training details of each of these models below. All models were trained across 5 random seeds and grid search was performed using the validation-set across all hyperparameters as mentioned below. Code is available at https://github.com/sinzlab/neural-sampling-neurips2023.

All computations were conducted on the shared high-performance compute cluster in the University of Washington (HYAK), which consisted of dedicated 8 x NVIDIA TESLA A100 GPUs and over 400 shared NVIDIA Turing and Ampere generation GPU's on a non-dedicated, as-available basis. We estimate end-to-end computation time for all models and experiments, including training and hyperparameter searches, to be roughly 3-4 days.

### E.1 Prior models

The prior models in our case are dequantization models (see paragraph under "Prior" in Section 2.2), which consist of a "continuous prior distribution", $p(\zeta; \omega)$ and a "dequantizer distribution", $q(\zeta|\mathbf{r}; \nu)$.

**Continuous prior** We consider three different distributions for $p(\zeta; \omega)$, namely: (1) Exponential distribution (**Exp**), (2) Half-normal distribution (**HN**) and (3) Normalizing flow-based distribution

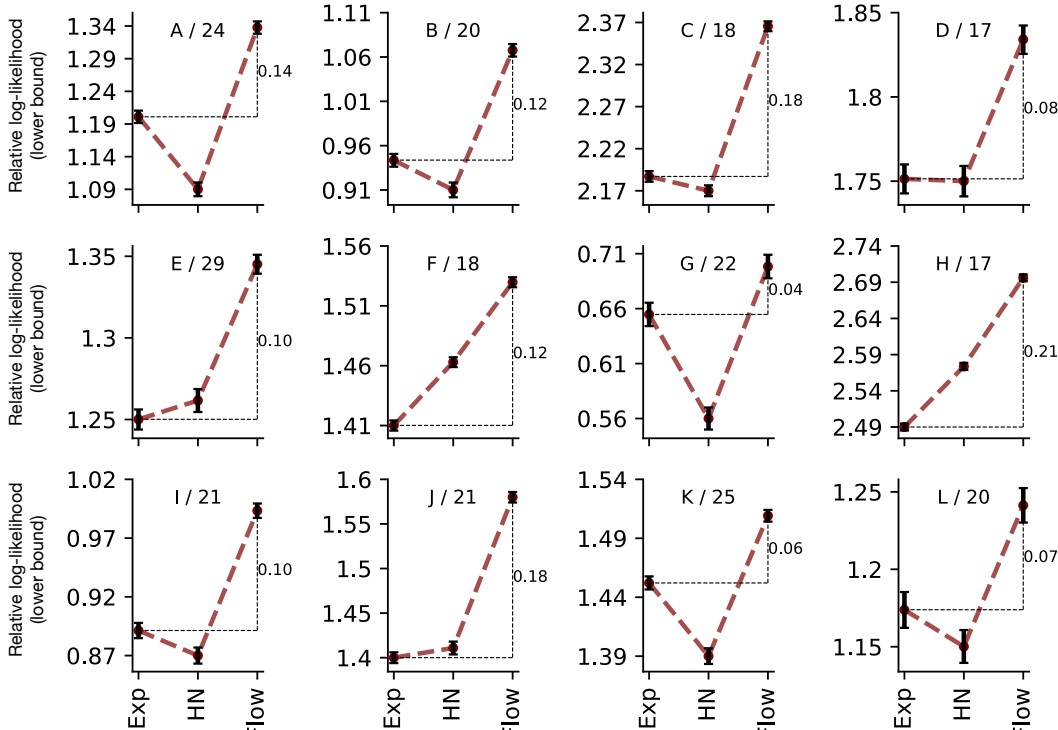

Figure 6: Relative log-likelihood scores (lower-bound), in bits of prior models, $p(\mathbf{r})$, on population recordings (test set), averaged by the number of neurons and trials. Each panel in the plot corresponds to prior models fit on one session. On top of each panel the label: "X/N" denotes "session-id/number of neurons". Refer to Table 2 for information on all sessions. The scores are computed relative to the Exp1 prior model baseline. We additionally denote the improvement the Flow model achieves, relative to the Exp model. Note that for the prior models on discrete spike counts, $\mathbf{r}$, we can only obtain a lower bound on $p(\mathbf{r})$. Here we show the importance-sampling bound (Equation (11)) with 1000 samples. Error-bars denote the standard error of mean computed across trials.

(**Flow**). Hyperparameter optimization was unnecessary for **Exp** and **HN** as they are standard distributions with only a single learnable parameter per dimension ($\lambda$ for **Exp** and $\sigma$ for **HN**). We used the following set of hyperparameters for the Flow model:

- Positive transformation: [`softplus`, `exp`, `square`, `ELU + 1`].
- Non-linearity after each affine layer: [`tanh`, `ELU`, `exp`, `log`].
- Learning rate: [1e-4, 5e-4, 1e-3]

**Dequantizer distribution** the dequantizer distribution is implemented via a conditional normalizing flow (Section B) and we used identical lists of hyperparameters as for the **Flow** continuous prior as listed above, and in addition considered the following hyperparameters w.r.t the conditioning function:

- Conditioning function: [`Linear`, `MLP`]
  - Nonlinearity for MLP conditioning function: [`RELU`, `tanh`]
- Weight-initialization for affine layers in the conditioning function: [$\mathcal{N}(0, 1e\text{-}3)$, $\mathcal{N}(0, 1e\text{-}5)$]
- Learning rate: [1e-4, 5e-4, 1e-3]

### E.2 Likelihood models

We model the likelihood as an isotropic Gaussian distribution

$$p\left(\mathbf{x}|\mathbf{r}^{(i)}\right) = \mathcal{N}\left(\mathbf{x}|\boldsymbol{\mu}^{(i)}, \boldsymbol{\sigma}^{2(i)} \cdot \mathbf{I}\right), \tag{13}$$

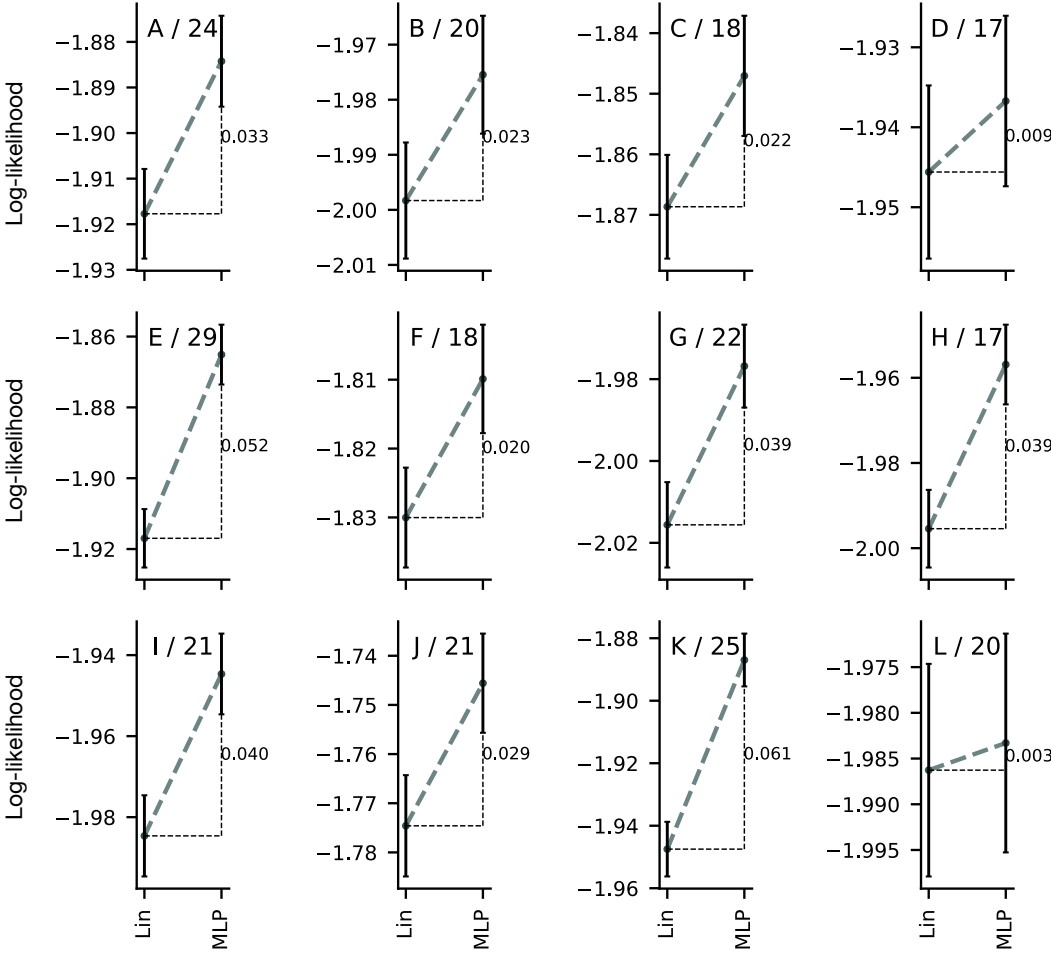

Figure 7: Likelihood scores of the likelihood models $p_{\mathbf{x}|\mathbf{z}}(\mathbf{x}|\mathbf{r})$, averaged across image pixels and trials. Each panel in the plot corresponds to likelihood models fit on one session. On top of each panel the label: "X/N" denotes "session-id/number of neurons". Refer to Table 2 for information on all sessions. We additionally denote the improvement the MLP model achieves, relative to the linear model. Error-bars denote the standard error of mean computed across trials.

where parameters mean, $\boldsymbol{\mu}^{(i)}$ and variance, $\boldsymbol{\sigma}^{2(i)}$ are functions of response, $\mathbf{r}^{(i)}$ and the function is either linear or nonlinear (see paragraph under "Likelihood" in Section 2.2). Operationally, both linear and nonlinear are formulated via an MLP (for linear, the MLP has no nonlinearities). We used the following sets of hyperparameters to train the MLP:

- Number of layers: $[2, \ 3, \ 4]$
- Nonlinearity: $[$RELU, Leaky RELU, none (linear)$]$
- Dropout rate: $[0, \ 0.5, \ 0.8]$
- Learning rate: $[1\text{e-}4, 1\text{e-}3]$
- Weight-initialization: $[\mathcal{N}(0, 1\text{e-}3), \ \mathcal{N}(0, 1\text{e-}5)]$
- L2 regularization strength: $[1\text{e-}1, \ 1\text{e-}3]$

### E.3   Posterior models

For each of the trained generative models, $p(\mathbf{x}, \mathbf{r}; \theta^*)$, we approximated the model's posterior distribution $p(\mathbf{r}|\mathbf{x}; \theta^*)$ using an approximate posterior $q(\mathbf{r}|\mathbf{x}; \phi)$ trained on samples from the trained generative model (Equation (10)).

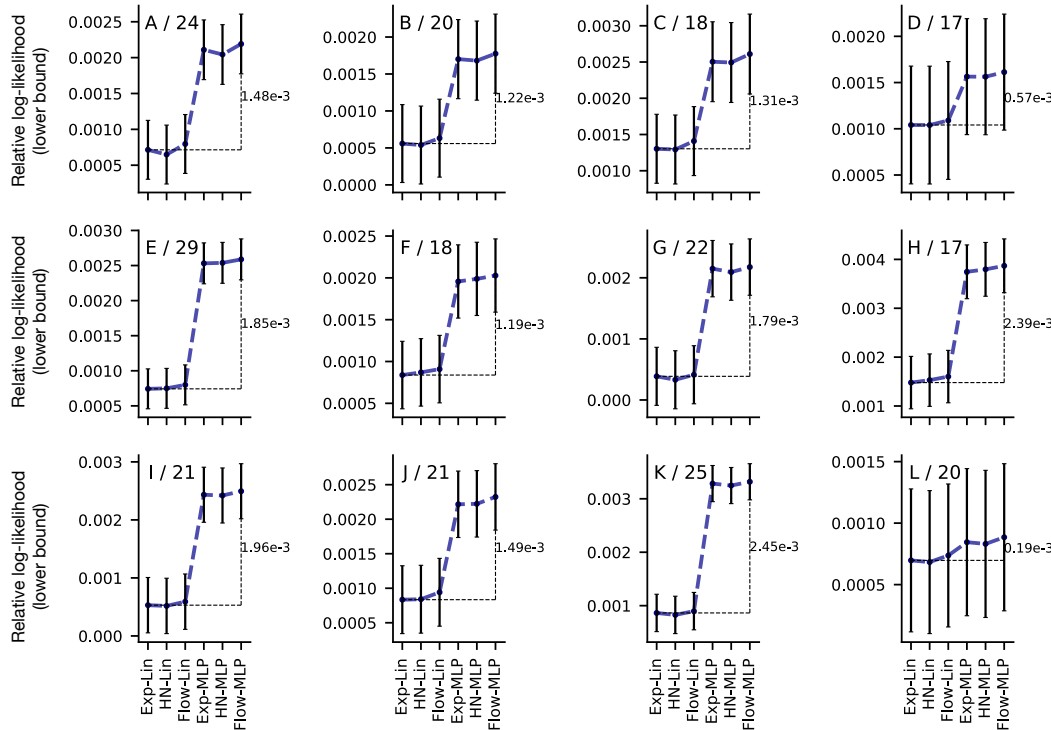

Figure 8: Relative log-likelihood scores (lower-bound), in bits of joint models, $p_{\mathbf{x},\mathbf{z}}(\mathbf{x}, \mathbf{r})$, on population recordings (test set), averaged by the number of neurons, image pixels and trials. Each panel in the plot corresponds to joint models fit on one session. On top of each panel the label: "X/N" denotes "session-id/number of neurons". Refer to Table 2 for information on all sessions. The scores are computed relative to the Exp1-Lin joint model baseline. We additionally denote the improvement the Flow-MLP model achieves, relative to the Exp-Lin model. Note that since for the prior models on discrete spike counts, $\mathbf{r}$, we can only obtain a lower bound on $p_z(\mathbf{r})$, the estimated joint log likelihood $\log p_{\mathbf{x},\mathbf{z}}(\mathbf{x}, \mathbf{r}) = \log p_{\mathbf{z}}(\mathbf{r}) + \log p_{\mathbf{z}}(\mathbf{r})$ is also a lower bound. Error-bars denote the standard error of mean computed across trials.

Since we have the trained generative model, we are free to sample as many pairs of neuronal responses and images ($\mathbf{x}', \mathbf{r}' \sim p(\mathbf{x}, \mathbf{r}; \theta^*)$) as we wish. We sampled 100,000 such pairs from each trained generative model and trained the approximate posterior on these samples respectively.

Note that the approximate posterior model maps images to the distribution over neuronal responses, and hence, we chose a nonlinear system identification model (Section C) architecture as its functional form.

The training details of the posterior models mirror that of the system identification model, except that posterior models are trained on samples from the trained generative model (Algorithm 1) and system identification models are trained directly on real responses to natural images. System identification model training is detailed below.

### E.4   System identification models

We trained the system identification models on monkey V1 data as described in Table 2, where we trained it separately for each session such that we can compare its performance to the NSC posterior models (Figure 5). We cropped the images to a size of 41x41 pixels and normalized them with respect to the mean and standard deviation of the images from the train and validation set. As objective function, we used the negative log-likelihood of a Gamma distribution whose parameters were predicted by the system identification model. The parameters of the system identification model itself were optimized using the Adam optimizer with an initial learning rate of 0.0042. This learning rate was reduced by 30% when the correlation between the mean of the predicted Gamma distribution

and the true responses on the validation set did not improve for three epochs. The training was stopped after reducing the learning rate three times.

# F   Details on analytical tractability of fitting literature models on simulated data

The classical models that we consider on simulated data (see "Simulated data" under Section 3), are ❶ a Hoyer & Hyvärinen model (HNH) [9], ❷ an Olshausen & Field (ONF) model [20], and ❸ a full Gaussian model (Gauss).

When fitting these classical models, the exact forms of the prior distributions in each of the models are as follows.

- HNH model: the prior is an exponential distribution : $p\left(\mathbf{r}_i\right) = \frac{1}{\lambda_i} \exp\left(-\lambda_i \mathbf{r}_i\right) H\left(\mathbf{r}_i\right)$, where $H$ is the heavyside function

- ONF model: the prior is a Laplace distribution, $p\left(\mathbf{r}_i\right) = \frac{1}{2b_i} \exp\left(-\frac{|\mathbf{r}_i - a_i|}{b_i}\right)$

- Gauss model: the prior is a simple isotropic Gaussian with mean $\mu_r$ and variance $\sigma_r^2$: $p\left(\mathbf{r}\right) = \mathcal{N}\left(\mathbf{r}|\mu_r, \sigma_r^2 \mathbf{I}\right)$.

All the three models share a common linear, isotropic Gaussian conditional distribution $p\left(\mathbf{x}|\mathbf{r}\right) = \mathcal{N}\left(\mathbf{x}|A\mathbf{r}, \sigma^2 \mathbf{I}\right)$.

Each of these (joint) models can be fit to simulated data analytically. This is possible since maximizing the joint distribution objective of each of the models (according to Equation (7)), boils down to maximum likelihood estimation (MLE) of standard distributions.

For HNH, the objective of maximizing the joint distribution would be:

$$
A^*, \sigma^*, \lambda^* = \underset{A,\sigma,\lambda}{\arg\max} \sum_{i=1}^{N} \overbrace{\log \mathcal{N}\left(\mathbf{x}^{(i)}|A\mathbf{r}^{(i)}, \sigma^2 \mathbf{I}\right)}^{\text{Likelihood } \log p\left(\mathbf{x}^{(i)}|\mathbf{r}^{(i)};\theta\right)} + \overbrace{\log \frac{1}{\lambda} \exp\left(-\frac{1}{\lambda}\mathbf{r}^{(i)}\right) H\left(\mathbf{r}^{(i)}\right)}^{\text{Prior } \log p\left(\mathbf{r}^{(i)};\theta\right)}.
$$

$$
= \underbrace{\underset{A,\sigma}{\arg\max} \sum_{i=1}^{N} \log \mathcal{N}\left(\mathbf{x}^{(i)}|A\mathbf{r}^{(i)}, \sigma^2 \mathbf{I}\right)}_{\text{MLE of conditional normal distribution}} ; \underbrace{\underset{\lambda}{\arg\max} \sum_{i=1}^{N} \log \frac{1}{\lambda} \exp\left(-\frac{1}{\lambda}\mathbf{r}^{(i)}\right) H\left(\mathbf{r}^{(i)}\right)}_{\text{MLE of exponential distribution}}
$$

$$(14)$$

Similarly, for ONF:

$$
A^*, \sigma^*, a^*, b^* = \underset{A,\sigma,a,b}{\arg\max} \sum_{i=1}^{N} \overbrace{\log \mathcal{N}\left(\mathbf{x}^{(i)}|A\mathbf{r}^{(i)}, \sigma^2 \mathbf{I}\right)}^{\text{Likelihood } \log p\left(\mathbf{x}^{(i)}|\mathbf{r}^{(i)};\theta\right)} + \overbrace{\log \frac{1}{2b} \exp\left(-\frac{|\mathbf{r}^{(i)} - a|}{b}\right)}^{\text{Prior } \log p\left(\mathbf{r}^{(i)};\theta\right)}.
$$

$$
= \underbrace{\underset{A,\sigma}{\arg\max} \sum_{i=1}^{N} \log \mathcal{N}\left(\mathbf{x}^{(i)}|A\mathbf{r}^{(i)}, \sigma^2 \mathbf{I}\right)}_{\text{MLE of conditional normal distribution}} ; \underbrace{\underset{a,b}{\arg\max} \sum_{i=1}^{N} \log \frac{1}{2b} \exp\left(-\frac{|\mathbf{r}^{(i)} - a|}{b}\right)}_{\text{MLE of laplace distribution}},
$$

$$(15)$$

Similarly, for Gauss:

$$
\begin{aligned}
A^*, \sigma^*, \mu_r^*, \sigma_r^* &= \underset{A,\sigma,\mu_r,\sigma_r}{\arg\max} \sum_{i=1}^{N} \overbrace{\log \mathcal{N}\left(\mathbf{x}^{(i)}|A\mathbf{r}^{(i)}, \sigma^2\mathbf{I}\right)}^{\text{Likelihood } \log p\left(\mathbf{x}^{(i)}|\mathbf{r}^{(i)};\theta\right)} + \overbrace{\log \mathcal{N}\left(\mathbf{r}|\mu_r, \sigma_r^2\mathbf{I}\right)}^{\text{Prior } \log p(\mathbf{r};\theta)} . \\
&= \underbrace{\underset{A,\sigma}{\arg\max} \sum_{i=1}^{N} \log \mathcal{N}\left(\mathbf{x}^{(i)}|A\mathbf{r}^{(i)}, \sigma^2\mathbf{I}\right)}_{\text{MLE of conditional normal distribution}}; \underbrace{\underset{\mu_r,\sigma_r}{\arg\max} \sum_{i=1}^{N} \log \mathcal{N}\left(\mathbf{r}^{(i)}|\mu_r, \sigma_r^2\mathbf{I}\right)}_{\text{MLE of normal distribution}},
\end{aligned}
\tag{16}
$$

## G  Evaluating Gamma-distribution on spike counts

Both the system identification model and the posterior model use a conditional Gamma distribution as the distribution of spike counts conditioned on images (see paragraph titled "Posterior" in Section 2.2).

Hence we compute the lower-bound using uniform dequantization ($K = 1,000$ samples) since we are evaluating our continuous Gamma-posterior on discrete spike counts, and full-likelihood is intractable [66–68]:

$$
\log P_{\text{lower-bound}}\left(\mathbf{r}|\mathbf{x}^{(i)}\right) \geq \log \mathbb{E}_{\mathbf{u}}\left[p_{\mathbf{z}|\mathbf{x}^{(i)}}\left(\mathbf{r} + \mathbf{u}|\mathbf{x}^{(i)}\right)\right] \approx \log \frac{1}{K} \sum_{k=1}^{K} p_{\mathbf{z}|\mathbf{x}^{(i)}}\left(\mathbf{r} + \mathbf{u}^{(k)}|\mathbf{x}^{(i)}\right)
$$

$$\tag{17}$$

where $\mathbf{u}^{(k)} \sim \mathcal{U}(0, 1)$.

Note that:

$$
p_{\mathbf{z}|\mathbf{x}^{(i)}}\left(\mathbf{r}|\mathbf{x}^{(i)}\right) = \prod_{j=1}^{S} p_\gamma\left(r_j|\boldsymbol{\alpha}^{(i)}, \boldsymbol{\beta}^{(i)}\right)
\tag{18}
$$

where $p_\gamma$ denotes the Gamma-distribution, and $j$ denotes the $j$th neuronal response.