# OpenReview forum: "Taking the neural sampling code very seriously: A data-driven approach for evaluating generative models of the visual system"
_NeurIPS.cc/2023/Conference — NeurIPS 2023 poster_

### Official Review · Reviewer_GaXp · 2023-06-23

**Soundness:** 2 fair
**Presentation:** 3 good
**Contribution:** 1 poor
**Rating:** 2
**Confidence:** 4

**Summary:**

The authors used several generative models to fit the image data and the corresponding neural responses. And for each generative model, they used the variational inference method to get their posterior.

**Strengths:**

The article is written in a clear manner.

**Weaknesses:**

The authors used several generative models to fit the image data and the corresponding neural responses. Neither the generative model nor the fitting method is biologically plausible. This cannot explain the working mechanisms of the neural system; it can only assess how well the model fits the data. And it is evident that more complex models tend to fit better. I fail to see the significance of this work.

**Questions:**

1. What do the two conditional probabilities in Eq.(1) mean? Could you provide a detailed explanation?
 When talking on NSC, I believe the latent variables z should be abandoned. Please see [1].

2. What does the term normative theory/assumption mean?

3. What insights can we gain from the the generative model fitted by neuronal population recordings?

Reference:
[1] Shivkumar, Sabyasachi, et al. "A probabilistic population code based on neural samples." Advances in neural information processing systems 31 (2018).

---

> ### Author Rebuttal · Authors · 2023-08-09
>
> We thank you for your critical feedback. We would like to address all of your points that you raise.
>
> **We would like to point out a misunderstanding that appears in your summary**. We do _not_ use variational inference in order to get the posterior, but rather we fit a model-posterior to the samples of the learned generative model (please see paragraph under “Posterior” in Section 2.2 “Models'' and Equation 9). We did so to ensure that we are not biased by the choice of approximation (i.e., ELBO). We realize this is a bit unconventional and can be misleading if not emphasized. We created an algorithm that summarizes our fitting methodology concisely and we state the posterior-learning part for your reference:
>
> Sample $\\{x'^{(i)}, r'^{(i)}\\}\_i^S \\sim p\_{x, z}(x, r; \\theta_P^*, \\theta_L^*)$\
> $\phi^* \gets argmax_\phi \sum_{i}^S \log q_{z|x}(r'^{(i)}|x'^{(i)}; \phi)$,\
> where $p\_{x, z}(x, r; \\theta_P^*, \\theta_L^*)$ is the learned generative model.
>
>
> **Re the meaning of the conditional probabilities in Eq 1**: In the manuscript, we let the subscript of the probability distribution designate distinct functions (probability density function for continuous and probability mass function for discrete random variables). Namely, $p_{r|x}$: probability distribution of $r$ conditioned on $x$, and $p_{z|x}$: probability distribution of $z$ conditioned on $x$. In Eq 1, $p_{r|x}(r|x)$ denotes the probability density function $p_{r|x}$ as evaluated on values of $r$ and $x$, and, $p_{z|x}(r|x)$ denotes the probability density function $p_{z|x}$ as evaluated on the values $r$ and $x$. Please also see our response to Rev **AuQ9**  for a reiterated version of our theoretical formulation.
>
> **Re latent variable $z$ in NSC**: We see your point about abandoning $z$ and replacing it with $r$ for NSC as is often done in the literature. We would like to highlight that such an approach entails an _implicit_ canonical assumption about the relationship between $r$ and $z$, specifically that there exists an equality in distribution between $r$ and $z$ (please also see our point under “Limitations I” under the “Discussion” section in the submitted manuscript). We deliberately use $z$ and $r$ as separate variables for two reasons: Firstly, our goal was to mathematically spell out this implicit equality assumption in NSC, and it was hence necessary to start with the idea that there exists a generative model of the world, and link neuronal responses to the latent variables in the generative model. Secondly, making the link explicit also allows one to formulate richer links between $r$ and $z$, which may be seen as extending NSC. For example, one could formulate that neuronal responses represent a subset of latent variables, involving nonlinear mappings. Ultimately, we believe that neuronal responses are a _reflection_ of the posterior samples. Being able to make these assumptions explicit and having a quantitative metric such as data log-likelihood performance allows one to see what assumptions describe the data best and aid follow-up experiments.
>
> **Re what normative theory/assumptions mean**: Normative theories aim to describe how a system _ought_ to function based on first principles, and involve assumptions with biophysical constraints (please see references [49] and [45] for a detailed exposition of normative theories). These principles often come in the form of optimality principles. A prominent example in visual neuroscience is sparse coding by Olshausen and Field (2004), where the underlying optimality principle is the maximization of energy efficiency and robustness to noise, among others. NSC itself is also a normative theory, postulating that the brain’s responses are samples from the posterior distribution over latent variables reflecting a generative model of the world and deriving neuronal coding properties from that. Commonly in NSC models, assumptions such as the equality of distributions between $r$ and $z$ (see point above) and the response/latent variables being sparse (hence modeled by an exponential distribution) are normative assumptions. In sharp contrast to system identification where the parameters of models are learned purely from data, normative theories often propose models that are either parameter-free or have parameters that are independent of data. Furthermore, normative theories are derived from broader computational frameworks, such as Bayesian inference, and have the ability to make predictions that are qualitatively different from system identification (such as how neuronal responses could encode uncertainty about the stimulus and adapt to changing stimulus distributions; please see our point in overall response, specifically “**Re 9, 10**”). Our work can be seen as an interpolation between system identification and normative modeling with the goal to make normative theory of NSC better testable on neurophysiological data.
>
> **Re biological implausibility of the fitting and models**: It was not our aim to specifically provide a biologically plausible model or algorithm. In the tradition of David Marr’s three levels, many prominent normative theories, such as the work by Olshausen and Field (2004), target the computational or representational level of what the computational goal of a system is and how stimuli are represented, but not how it is implemented in a neuronal system (implementational level). Our work targets models for these levels. Asking how a neuronal system would implement such a model and how it would learn it, is an interesting and scientifically important question, but adds a lot more complexity which would require a dedicated treatment. For that reason, we did not focus on that.
>
> **Re what insights can be gained by fitting generative models to data**: Please refer to our overall response (specifically “**Re 9, 10**”).

---

> > ### Comment · Reviewer_GaXp · 2023-08-11
> >
> > **My primary concerns have not yet been addressed. I don't believe this approach test the normative theory of NSC.** When demonstrating a normative theory, the implementational level cannot be disregarded; otherwise, the focus would be solely on validating the quality of data-fitting models. The current work involves fitting various generative models to experimental data. In practice, you can propose a theory, design a model accordingly, and then use data to fit this model. Here's an example:
> > 1. normative theory: the nerual response $r$ is determined by the weather $x$.
> > 2. data:  $x$: temperature distribution, air humidity distribution..., $r$: neural reponse.
> > 3. model: $y=f1_\theta(x)$,$y = f2_\theta(x),...$, where $f1_\theta(\cdot)$ and $f2_\theta(\cdot)$ are deep neural networks.
> > 4. fitting method: Any optimization algorithm.
> > 5. result: I provide a method which can test my normative theory on weather and neural response.
> >
> > Clearly, such a pipeline approach is nonsensical as it fails to account for how weather influences neural response at the implementation level.
> >
> > **I didn't make a misunderstanding of the approach in approximating posterior.** You are not familiar with variation inference (VI). Actually, in VI, a tractable distribution $q_\phi(r|x)$ is applied to approximating the true posterior $p(r|x)$, writting as $min_\phi E_{x\sim p(x)}D_{KL}(p(r|x)||q_\phi(r|x))$, which is equivlent to $max_\phi E_{r,x\sim p(x,r)}q_\phi(r|x)$, which is just Eq.(9) in your main text.
> >
> > **You didn't eliminate my concern about $z$.** In your reply to reviewer AuQ9 on PPC, you mentioned that 'experimenter-defined variable (orientation) is different from the latent variable $z$ you used in your main text'. However, a gabor-liked orientation filter in Figure 1A illustrating $z$ can be really confusing. What you are saying and writing seem to be contradictory.
> >
> > **I'd like to reiterate once again that the presence of $z$ is not necessary.** Indeed, according to your definition, $z$ is not utilized in the fit model to evaluate the model, and its role is entirely equivalent to $r$. Actually, you have a misunderstanding of NSC. In the framework of NSC, the generative model is an intrinsic model within the neural system where $r$ is the latent variable and $x$ is the obvservation. The variable $z$ is an imagined, fundamentally non-existent entity, as it can never be directly observed. This aligns with its designation as a latent variable rather than an observable variable.
> >
> > **Where is reference [49]?**

---

> > > ### Author Response · Authors · 2023-08-13
> > >
> > > Thanks for your prompt response!
> > >
> > > **Re reference [49]**: “49” was a typo, we are sorry for the confusion. We mean [46] in the main text, namely: Levenstein et al. “On the role of theory and modeling in neuroscience” (2023).
> > >
> > > **Re primary concern**: Again, we would like to respectfully disagree with you. Going by Levenstein et al. (2023), neither normative theories nor descriptive theories (aka phenomenological theories, such as system identification), unlike mechanistic theories, are necessarily obligated to consider implementational details and propose biologically plausible models. Our work can be viewed as an interpolation between the normative theory of NSC and phenomenological system identification. Many well known normative theories do not provide implementational details in their demonstration. For instance, Olshausen and Field (1996) use a sparse linear overcomplete model, and Hoyer and Hyvärinen (2002) use a modified version of the same. Neither of them provide details of how a network of spiking excitatory and inhibitory neurons would implement these models. In fact, we fit the Hoyer and Hyvärinen model in our framework, so our models use the same level of implementational detail as existing models.  That said, we agree with you that normative theories should introduce restrictions on the structure of the models and not just fit any flexible model. This is indeed the case in our work: for example, our models are derived from formalizing NSC, and we use linear likelihood (as in Hoyer and Hyvärinen (2002)), sparse prior and prior independence of neurons (Hoyer and Hyvärinen (2002) and Olshausen and Field (1996)), and show how they describe data (likelihood).
> > >
> > > **Re variational inference**: We realize there was a misunderstanding. We agree that “using a tractable distribution and optimization in order to approximate the posterior distribution” is the essence of variational inference. What we wanted to clarify was that we do not use _ELBO-based variational inference_, where the objective for the posterior is $min_\\phi D_{KL}(q_\\phi||p)$ (note the order of $p$ and $q$ is different from what you wrote). What we do is $max\_\\phi E\_{x, r \sim p_\{\theta_p^*, \theta_L^*}(x, r)} q\_\\phi (r|x)$, which is equivalent to $min_\\phi E\_{x\sim p\_{\\theta^*}} D_{KL}(p\_{\\theta^*} || q\_\\phi)$.
> > >
> > > **Re $z$ and PPC**: In PPC, the experimenter-defined variable of orientation is not a latent variable, but a task variable. One could compute a posterior over the task variable and decode it from neuronal responses as long as the responses carry useful information in performing decision making as part of the task. But that does not imply that the neuronal responses are samples from that posterior distribution (NSC). The latent variable that the neuronal responses could be representing can be different from the task variable. Hence, orientation is a possible latent variable but it does not have to be. Coming to oriented grating that we depict in Fig 1A: that was done for illustrative purposes as to what the latent variable could potentially be. We will make sure to clarify this distinction in the main text so as to not cause confusion.
> > >
> > > **Re necessity of $z$**: We agree with you that the generative model is internal to the neural system, and “$z$” is not observable, and therefore we also call it the “_latent_ variable”. However, multiple papers on NSC qualitatively compare the latent variable of the normative model with neuronal responses properties, implying that there is a functional relationship. Our goal is to make this relationship quantifiable for single recorded neurons. We would therefore like to reiterate why we use $z$: in the original formulation by Hoyer and Hyvärinen (2002), the neuronal response itself represents the latent variable. Thus, once neuronal responses are observed, the latent variable actually becomes unnecessary/effectively observed. However, since then the NSC literature has interpreted neuronal responses under NSC in different ways. For example, in Orban et al (2016), membrane potential values (neuronal responses), are modeled as a nonlinear function of the samples over the posterior distribution of feature activations (latent variable) that generates the image (stimulus). In this case, there is a distinction between the latent variable $z$ of the generative model and neuronal responses $r$ ($z$ are $r$ are related by a nonlinear function). Similarly, in Savin and Denève (2014), responses of $N$ neurons relate to a $D$ latent variables where $N > D$.  Many more ways of how $r$ and $z$ relate are conceivable. Our formulation with a separate $z$ and $r$ makes it in principle also applicable to these cases, which is the reason we decided to present it this way. In this work, we wanted to focus on the aspect of fitting these models to data. Given this, we chose the simplest (original) interpretation of NSC where $r=z$. We will emphasize this further point in our main text.

---

> > > > ### Comment · Reviewer_GaXp · 2023-08-15
> > > >
> > > > **Primary concern** I stand by my original viewpoint: bold assumptions should be made while being cautious in seeking evidence to support them.
> > > >
> > > > If you're proposing an entirely new normative theory, implementation details might not be a primary concern at the outset. When a new theory is introduced, the focus is often on its novelty and the fresh insights it can offer. More potential theories usually diverge more from existing ones, leading to a need for further details. However, when proving a normative theory, especially during its implementation, extreme care is necessary, and it cannot be overlooked. Even though you have imposed certain constraints on your model, they may not be sufficient for thorough proof.
> > > >
> > > > Indeed, we have differing perspectives when it comes to the validation of normative theories. However, I hold the belief that my viewpoint is more conducive to the advancement of science.
> > > >
> > > > Returning to the examples you mentioned, Hoyer and Hyvärinen (2002) are the proponents of the NSC theory, so their model's lack of biological plausibility is understandable. Olshausen and Field (1996), while their model also lacks biological realism, explained the receptive field phenomenon and trained Gabor-like receptive fields. Does your model, aside from fitting the data, reveal any biological phenomena? Does it provide additional insights?

---

> > > > > ### Author Response · Authors · 2023-08-17
> > > > >
> > > > > **Re primary concern**: The main focus  of this work is to develop a way for answering the question, “How well does NSC explain neurophysiological data _quantitatively_?”. Since even normative models have remaining degrees of freedom (such as a linear filter or a choice of nonlinearity), can we infer those from data? The primary contribution of the paper is hence a way to formulate NSC such that it allows us to develop the methodology to approach these questions. While we share your curiosity towards feasible implementational aspects of the theory, we believe that a data-driven quantitative approach has merits on its own, independent of implementational aspects.
> > > > >
> > > > > The seminal work by Olshausen and Field is in fact a good motivation for our work. Olshausen and Field train a purely image-based model. Since their model is trained independently from any neuronal data, it is hard to compare how well it actually explains neuronal response properties in the visual cortex. One way to do this is to compare learned receptive fields (RFs) to those of real neurons [1]. However, since neurons are shown to be complex nonlinear functions of stimuli, they are more than their RFs. In addition, many models and objective functions yield Gabor filters. Which one is a better model for the visual system? Ideally, we would like a method to check how a normative model like that of Olshausen and Field fits the real neuronal responses and quantitatively compare it to alternative models. This is what we strived to achieve in the submitted work.
> > > > >
> > > > > **Re additional biological insights**: Here, we neither intended nor claimed to provide any additional biological insight. We do not offer a new normative theory, and the flexible models that we develop are best seen as straightforward extensions to classical models. Critically, what we provide is a methodology for quantitative model comparison of models under NSC. By doing so, we show that traditional models of NSC actually do not predict neuronal data well, especially compared to a flexible system identification model. This suggests that the assumptions in these normative models may be too restrictive and need to be modified to provide a better explanation for neuronal data. One important aspect of scientific progress is to make things measureable. This is what we provide here.
> > > > >
> > > > > Additionally, in our overall response  (specifically "**Re 9, 10**"), we describe an experimental test that our formulation of NSC and fitting framework lets us conduct. We believe this is a biologically relevant theoretical insight that our work provides us. Core to the experiment is the idea of using the learned generative models to make predictions for sensory neuronal adaptations to changed sensory contexts. We also believe this would be a strong test of the theory of NSC itself (please also see "**Re confirmation or refutation of NSC**" in our response to Reviewer 1S3S).
> > > > >
> > > > > [1] van Hateren JH and van der Schaaf A. Independent component filters of natural images compared with simple cells in primary visual cortex. Proc R Soc Lond B Biol Sci 265: 359-366, 1998.

---

### Official Review · Reviewer_D2be · 2023-07-04

**Soundness:** 3 good
**Presentation:** 3 good
**Contribution:** 3 good
**Rating:** 6
**Confidence:** 4

**Summary:**

The authors propose a novel formalization of the Neural Sampling Code (NSC) in this paper. The authors aim to bridge the gap in existing NSC research that needs more quantitative alignment between normative theory and neuronal recordings.

The authors propose formalization enables fitting NSC generative models directly to neuronal activity, creating richer, more flexible models and allowing standard metrics to evaluate different generative models.

The authors propose to compare classical and deep learning-based generative models on population recordings in V1 in response to natural images. Their framework outperforms classical models in both generative- and predictive-model performance. They also propose a method for obtaining stimulus-conditioned predictive models of neuronal responses that can be compared to existing neural system identification models.


**Strengths:**

The authors propose a novel formalization of the Neural Sampling Code (NSC), allowing for the direct fitting of NSC generative models to recorded neuronal activity. Also, the paper presents a new way to evaluate different generative models under NSC using standard metrics quantitatively.

The study compares classical and deep learning-based generative models, providing valuable insights into the capabilities of different generative models in capturing the structure of real neuronal responses.

The research is grounded in the practical application of the theory, using neuronal activity data from a macaque primary visual cortex (V1) in response to natural images.

Finally, the authors spend a significant amount of space identifying the limitations and suggesting areas for future research, promoting ongoing dialogue and exploration in the field.


**Weaknesses:**

I believe that the main weaknesses of the current paper have been presented by the authors in the discussion section but do not undermine the validity of the current study.

However, one important limitation lies in the experimental section. The paper provides limiting experimental results when it comes to different neural recordings. In order to clearly evaluate the validity of the claim, one would need to explore other brain areas, probably still in the early sensory processing regions.

**Questions:**

How does the model perform in other brain regions?

**Limitations:**

Clearly laid out by the authors

---

> ### Author Rebuttal · Authors · 2023-08-09
>
> We thank you very much for your supportive and motivating review. We hope that our work promotes the ongoing dialogue and exploration in the field.
>
> **Regarding different brain regions**: We share your interest and curiosity in exploring and testing our approach comprehensively in different brain regions and also across different animal species. Given that we have our current framework in place, we are excited that it gives us the opportunity to do so in an upcoming study.
> However, the focus of our current work was to establish a novel formulation of NSC that facilitates fitting them to recorded neurophysiological data. Testing it on different brain regions, which likely have no established normative models yet, was out of the scope for the current study. We chose the macaque primary visual cortex (area V1) because most of existing NSC literature has focused on modeling the V1 population responses to visual stimuli. We would like to emphasize that our approach is indeed not limited to V1, but serves as a broader framework that can be used to develop and fit models in other brain areas.
>
> The consistency of results across all tested datasets gives us confidence in the reliability of our approach and its potential for generalizability of findings in other brain regions.

---

> > ### Comment · Reviewer_D2be · 2023-08-16
> > **Response to the authors**
> >
> > I thank the authors for their response.
> >
> > I have also read the comments and reviews of the author reviewers. I believe there is some value in the current work, but I can see the limits pointed out by other reviewers. I will keep my current rating.

---

### Official Review · Reviewer_1S3S · 2023-07-06

**Soundness:** 3 good
**Presentation:** 3 good
**Contribution:** 3 good
**Rating:** 6
**Confidence:** 4

**Summary:**

The authors employed a generative model approach to predict V1 neural responses, aiming to evaluate the Neural Sampling Code (NSC) using a quantitative metric. The primary potential contribution lies in the theoretical formulation and the idea that neural response prediction can be utilized to assess the NSC hypothesis, especially on a per-trial basis.



**Strengths:**

The authors employed a generative model approach to predict V1 neural responses. This is a novel contribution in the area of single neuron response prediction, even though generative models like VAE, beta-VAE, and hierarchical VAE have been commonly used in analyzing fMRI data, particularly for reconstructing the input image.

The primary potential contribution lies in the theoretical formulation and the idea that neural response prediction can be utilized to assess the NSC hypothesis, especially on a per-trial basis.  The proposal that the performance of a  generative model for predicting neural response can provide a quantitative metric for evaluating the Neural Sampling Code (NSC) is new and interesting, even though I have not been completely convinced by this work presented (see below).  Nevertheless,  I put this paper as a "marginal accept" for the idea and the theoretical contribution.

Novel and substantial macaque V1 data.

**Weaknesses:**

The authors employed a generative model approach to predict V1 neural responses, aiming to evaluate the Neural Sampling Code (NSC) using a quantitative metric.   However, the response prediction performance, as measured by correlation, was  poor compared to the standard state-of-the-art feedforward neural network system identification model. Hence, the virtue and the effectiveness of utilizing a generative model in this context remain unclear.

More importantly,  it is unclear how this work confirms or refutes the NSC hypothesis. Moreover,  the earlier conceptual framework and theoretical formulation of the NSC propose that samples of population activities within a 10-20 msec window are used to approximate the posterior via Markov chain Monte Carlo (MCMC). In contrast, this work uses responses within a 120 msec duration throughout the entire stimulus presentation as a sample. Consequently, I find it hard to relate the current results to the original NSC idea. Additional arguments and explanations are necessary to establish the connections more convincingly.



**Questions:**


How does the current work speak for NSC hypothesis? Does the result in this paper confirm or refute it?


**Limitations:**

No discussion of societal impact.

---

> ### Author Rebuttal · Authors · 2023-08-09
>
> We thank you for your encouraging review.
>
> We are positive that we can address all your questions below.
>
> **Re the correlation performance gap between the posterior and system identification**: Please see our overall response (specifically “**Re 9, 10**”)
>
> **Re confirmation or refutation of NSC**: Our work can be used to (1) learn generative models directly from neurophysiological population recordings and natural images and to (2) subsequently quantitatively compare different generative models proposed for NSC. These comparisons can then form a solid basis to compare different normative assumptions, realized via specific choices in the generative model specification such as sparsity realized via exponential distribution, but all under NSC. Hence, our work in itself neither refutes nor confirms NSC. Rather, (1) we envision future work to use our framework for quantitatively comparing NSC models to models derived from other theories, but most crucially (2) if NSC were true, i.e responses are samples from the posterior distribution $r \sim p(z|x) \propto p(x|z)p(z)$, then changes in $p(z)$ would be reflected in the responses. We believe that designing experiments that aim to test this idea would be a strong way to arrive at confirmation or refutation of NSC, and our framework that lets us learn the generative model would serve as the necessary prerequisite. For a more detailed discussion, please refer to our overall response (**Re 9, 10**).
>
> **Re choosing “sample” as total spike count over 120 ms different from literature**: We would like to point out that, as far as we know, there is no generally agreed upon or a rigorous definition of a “sample” in NSC. While NSC was originally motivated with firing rate/spike count over a 500ms window as the sample (Hoyer and Hyvärinen, 2002 referring to A. F. Dean. 1981), some works even employed membrane potential over 10ms (Orban et al. 2016) as the definition of a sample. It is unclear on what generally applicable metric such a definition could be evaluated, _other than goodness of fit to data_. In this work, we chose the total spike count as the working definition. However, identifying the right sample definition falls within the scope of modeling and fitting NSC models to the data. Now that we can quantitatively fit neuronal responses under NSC, we could evaluate different definitions of samples to test which leads to the highest likelihood under  NSC. To us, this was another motivating factor for striving towards a data-driven evaluation of sampling models that allow us to compare these choices in an informed and quantitative way.

---

> > ### Comment · Reviewer_1S3S · 2023-08-20
> > **Thanks for the responses.**
> >
> > I have read the authors' responses which did address my questions. I agreed that the work could provide a valuable framework for evaluating NSC models or hypothesis in the future.  Given the sampling hypothesis popularized recently by Orban et al typically used a sampling window of 10 ms, it would be interesting to compare how well the different sampling window sizes would work. The poor performance of the generative model compared to a simple CNN remains a concern.

---

### Official Review · Reviewer_AuQ9 · 2023-07-06

**Soundness:** 4 excellent
**Presentation:** 2 fair
**Contribution:** 4 excellent
**Rating:** 6
**Confidence:** 4

**Summary:**

This paper proposes a statistical framework to infer the generative models hold in the brain from population activity. This study first analytically proves that it is feasible to infer the generative model from population activity. Then, the study compares various combinations of prior and likelihood to fit synthetic and empirical data. They found that the flow-MLP model outperformed other models. In particular, the flow-MLP model allows to flexibly capture the true generative models.

**Strengths:**

Overall I am positive about this study. This study is well motivated, interesting, and technically sound. The discussion part is also inspiring.

1. Neural sampling codes (NSC) is an important theory of population activity in the brain. This study definitely provides a practical framework that neural sampling codes can be useful in analyzing high-dimensional neural data.
2. Using population activity to infer generative models is an excellent idea. This study is well motivated.
3. The experiment is complete and neat. The study includes three priors x two likelihood functions, as well as a system identification model for comparison.

**Weaknesses:**

1. The writing can be improved. The overall texts are clear but some mathematical notations lack explicit explanations.
2. The normalizing flow model needs more detail.
3. More explanations are needed about in what situation this approach is useful in future empricial experiments. Or how experimentalistscan  design and perform better experiments to promote this line of research.

**Questions:**

1. The logic of neural sampling codes is very clear. I am not doing this line of research but I have know  the difference between neural sampling codes (NSC) and probablistic population codes (PPC) for a long time. More insights are needed to compare NSC and PPC. Is it also feasible to infer generative models under the framework of PPC. More discussions are needed.
2. I am really confused by several mathematical notations here.
   * In Eq1, I understand the meaning of r, x, z. but the footnote $r|x$ and $z|x$ are unclear to me. Given this is the key equation in this paper.
   * Eq2, "This subtle and seemingly trivial match implies that there exists a one-to-one mapping between the  space of neuronal responses and the latent variable underlying the stimulus". This sentence is very confusing to me.
   * Lines85-87, "However, intriguingly, our formalization in equation 1 allows us to directly learn the generative model pz,x (z, x) by instead learning the probabilistic mapping between the neural responses and the stimuli". I think this is they key logic. However, given the mathematical notations in Eq1&2 are confusing. This logic is really hard to follow.
   * In Eqs.6&7, what are the $w$ and $v$? I assume they are parameters of the continous latent distribution? Better to clarify.
3. In what situations is this approach useful. Can the authors give some cases where experimentalists can apply this approach to their data, or can design a better experiment to test relevant theories.

**Limitations:**

This study overall is well done. Some notations should be better explained and more discussions are needed in terms of relevance to empirical experiments.

---

> ### Author Rebuttal · Authors · 2023-08-09
>
> We thank you for your positive and supportive review. We are positive that we can address all your comments and questions below.
>
> **Re clarification of notation and meaning behind Eq 1, 2 and 3 (key logic)**: In the manuscript, we let the subscript of the probability distribution designate distinct functions (probability density function for continuous and probability mass function for discrete random variables). Namely, $p_{r|x}$: probability distribution of $r$ conditioned on $x$, and $p_{z|x}$: probability distribution of $z$ conditioned on $x$. In Eq 1, $p_{r|x}(r|x)$ denotes the probability density function $p_{r|x}$ as evaluated on values of $r$ and $x$, and, $p_{z|x}(r|x)$ denotes the probability density function $p_{z|x}$ as evaluated on the values $r$ and $x$.
> In its simplest form, NSC posits that _a neuronal response to a stimulus is a "sample" from the posterior distribution of latent variables given the stimulus_, which we mathematically write as: $ z_{sample} \sim p_{z|x}(z|x)$ and $r = I(z_{sample})$, where $I$ is the identity function.  What this implies is that there exists a one-to-one correspondence between the $r$ and $z$. In other words, $r$ and $z$ are two random variables that share the same (1) stimulus-conditioned distribution (Eq 1) and (2) marginal distribution (Eq 2). Consequently, the equivalence allows for the generative model of the stimulus $p_{x, z}(x, z) = p_{x|z}(x|z) p_z(z)$ to be learned directly using stimuli $x$ and responses $r$, where $r$ can be used in place of $z$ (Eq 3).
>
> **Re details on normalizing flow**: For modeling the prior distribution over spike counts $P(r)$, we used a normalizing flow distribution to model the continuous prior $p(\zeta)$ over the dequantized neural responses $\zeta$ (see Equation 5).
> We assumed that prior distributions over each neuron response are independent and thus can be factorized: $\log P(r) = \sum_j^n \log P(r_j)$, where $j$ is the neuron index over $n$ neurons.
> The same independence assumption was applied for the continuous prior for the dequantized neuronal responses  $\zeta$.
> We will now describe the model for a single dimension $\zeta_j$, and for simplicity, we drop the subscript $j$.
>
> We rephrase our normalizing flow distribution in two steps:
> $p({\zeta};\omega)=p_{\text{base}}(T^{-1}({\zeta};\omega))\cdot|\frac{\partial T^{-1}({\zeta};\omega)}{\partial {\zeta}}| = \mathcal{N}(T^{-1}({\zeta}; \omega)|0,1)\cdot |\frac{\partial T^{-1}({\zeta}; \omega)}{\partial {\zeta}}|$
> where we choose $p_{\text{base}}$ to be $\mathcal{N}(\cdot|0, 1)$, a standard normal distribution, and $T^{-1}$ represents the following (ordered) series of invertible mappings with learnable parameters $\omega$: $[\text{affine}, \tanh, \text{affine}, \tanh, \text{affine}, \tanh, \text{affine}, \text{softplus}^{-1}]$, where $\text{softplus}^{-1}(y) = \log(e^y - 1)$, $\text{affine}(y) = ay + b$ with learnable parameters $a$ and $b$. $\text{softplus}^{-1}$ ensures that the support of ${\zeta}$ is non-negative, reflecting (non-negative) spike counts.
>
> We identified two errors in our normalizing flow description in the manuscript: (1) we wrote the $\text{softplus}^{-1}$ layer at the wrong end of the transformation (Line 144) and (2) we used $\xi$ as the parameters of the flow whereas it should have been $\omega$ (Lines 125, 144). We will correct them both in the main text.
>
> **Re symbols in Eqs 6 & 7**: We thank you for bringing this to our attention. We missed the description of these two parameters, and will add them to our main text. $\omega$ refers to the parameters of the continuous latent distribution modeled by the normalizing flow $p(\zeta; \omega)$, and $\nu$ refers to the parameters of the dequantization distribution $q(\zeta|r; \nu)$.
>
> **Re feasibility of inferring generative models under PPC**: We thank you for prompting a discussion on PPC and we will add the following discussion to our main text. In its original formulation (Ma, et al 2006; Beck et al 2008), PPC focuses on capturing the likelihood function $\mathcal{L}(x) \equiv p(r|x)$, relating the population response to some specific experimenter-defined quantities such as orientation. Hence, fitting PPC models to neural responses amounts to capturing only this likelihood function and contains no notion of the prior over the stimulus $p(x)$. We hence believe that PPC does not readily offer a means for fitting a generative model, complete with the prior. That being said, there exist extensions to PPC where the responses are seen as encoding a posterior distribution of an experimenter-defined variable (e.g. orientation) via a specific distributional family (i.e. exponential family) (Shivkumar et al. 2018). However, in such work, the models are typically limited in that it only captures the relationship between the population response and some experimenter-defined variables such as stimulus orientation. This is in contrast to NSC in which latent variables are seen as what underlies the (natural) stimulus but not necessarily any experimenter-defined variable. To our knowledge, there currently exists no method to identify/learn the target latent variable z and the generative model over (z, x) based on the observed population responses r within the framework of PPC. While this is certainly an interesting avenue of research, deriving such a method would require dedicated work, perhaps similar in extent to our current work on the NSC.
>
> **Re relevance to experiment**: Please see overall response (“**Re 9, 10**”).

---

> > ### Comment · Reviewer_AuQ9 · 2023-08-19
> >
> > Thanks for your reply. Some writing problems are solved.
> >
> > I read comments from other reviewers. I kind of agree with you that, at least in this problem context, the implementation level is not that important. I think what you did here is at the same level as Hoyer and Hyvärinen (2002) and Olshausen and Field (1996).
> >
> > However, I noticed that Reviewer GaXp mentioned the usage of $z$. I strongly recommend you discuss what are exactly the latent variable, the observable variable here, and those in Hoyer and Hyvärinen (2002) and Olshausen and Field (1996), and their differences and relations. Hoyer and Hyvärinen (2002) describes it as "world parameters". Is this your latent variable here?? These discussions are good for audiences to better capture your idea.
> >
> > I cannot comment on these definitions, but I decided to keep my score here.

---

### Author Rebuttal · Authors · 2023-08-09

We thank the reviewers for the valuable feedback on our manuscript. We are happy that they found our work to be “well-motivated, interesting and technically sound” (Rev **AuQ9**), with “novel contribution in single response prediction” (Rev **1S3S**), “grounded in practical application of the theory” (Rev **D2be**) and providing “a practical framework that NSC can be useful in analyzing high-dimensional neural data” (Rev **AuQ9**). They remarked on our experiments as “complete and neat” (Rev **AuQ9**) and that we test on “substantial macaque V1 data” (Rev **1S3S**), and found our discussion to be “inspiring” (Rev **AuQ9**).

The reviewers also raised several points to be addressed:

**1**. To clarify our notation, some probabilistic objects (Rev **AuQ9**, Rev **GaXp**) and the logic behind our formulation (Rev **AuQ9**).
**2**. To add details for the normalizing flow (Rev **AuQ9**).
**3**. What normative theory/assumption means (Rev **GaXp**).
**4**. To abandon latent variable $z$ (Rev **GaXp**).
**5**. On biological implausibility of our methodology (Rev **GaXp**).
**6**. To discuss our “sample” definition as spike count over 120ms different from literature (Rev **1S3S**).
**7**. On feasibility of inferring generative models under PPC and make connections (Rev **AuQ9**).
**8**. How models would perform in other brain regions (Rev **D2be**).
**9**. What the benefit is from learning a generative model on responses and stimuli (Rev **GaXp**), especially when the posterior correlation is lower compared to system identification (Rev **1S3S**).
**10**. What new experiments could be designed from using this approach on data, in order to test relevant theories (Rev **AuQ9**), and
**11**. Whether our study confirms or refutes NSC (Rev **1S3S**).

We are confident that we can address all of the points in our responses. In this overall response, we first summarize the **main intended contribution** of the paper, since it is relevant for all the questions. Then we  respond to points **9** and **10** since we believe it carries significance relevant to all reviewers. Other points (**1 - 8, 11**) are answered in the responses to the individual reviewers.

**Summary**: This work is a contribution to the idea of the Neural Sampling Code (NSC) in computational neuroscience. While it is a prominent theory, there exists no work that formulates it such that NSC models can be fit to recorded neurophysiological data. The central goal of our current work is to (1) show how to mathematically express NSC which allows us to (2) fit generative models under NSC to neural responses to natural images, and (3) quantitatively compare different models with standard metrics such as data-likelihood. This allowed us to develop and fit deep learning-based generative models that are more flexible than existing literature models, and they outperform them when applied to recorded V1 responses to natural images. We also show (4) that we can derive a response-predictive model from NSC using the learned generative model’s posterior, which can be directly compared to system identificatio. Equipped with this framework, we (5) show that NSC models are not powerful enough to account for V1 responses to natural images compared to system identification. This motivates future work that can close the gap between normative NSC theory and data driven system identification.

**Re 9, 10**: Currently, system identification models are the state-of-the-art in predicting responses to natural images. To our knowledge, our work is the first to show that generative models under NSC fail to predict neuronal responses well compared to system identification models. There is still much room to build better generative models which would improve the posterior performance and our work builds the quantitative foundation for it.

But, for a _given dataset_ of responses and stimuli, we do _not_ expect the posterior from even the _true_ generative model to surpass the performance of the _best_ system identification model, because the generative model does not provide any advantage over system identification in response prediction, unless some specific inductive biases were to be introduced in the generative model.

However, if we change the stimulus distribution $p(x)$ to $p\_{new}(x)$ with markedly different stimulus statistics and let the sensory neurons adapt to $p\_{new}(x)$, we would expect the system identification performance to drop on $p\_{new}(x)$ and the system identification model might have to be retrained on a new dataset of responses under $p\_{new}(x)$.

This is the case where we could expect the generative model to be beneficial. Specifically, change in $p(x)$ to $p_{new}(x)$ may entirely derive from the change in prior $p(z)$ to $p_{new}(z)$, while $p(x|z)$ remains fixed, since $p(x|z)$ hypothetically represents the invariant “physical” process by which the latents (e.g., identity of an animal) give rise to observations (e.g., appearance of the animal). Consequently, if NSC were to be true i.e, $r \sim p(z|x) \propto p(x|z)p(z)$, the neuronal adaptation can be accounted for via simply learning $p_{new}(z)$, i.e., $r_{new} \sim p_{new}(z|x) \propto p(x|z)p_{new}(z)$, keeping $p(x|z)$ fixed. Importantly, $p_{new}(z)$ can be learned from the new stimulus samples alone from $p_{new}(x)$ _without the need of a new dataset of responses to the new stimulus samples_.

This insight helps us identify potential future experiments to rigorously test NSC models utilizing our framework since our it lets us learn the generative model $p(x|z)p(z)$ via $p(x|r)p(r)$ (NSC assumption). Namely, we would perform experiments in which we let the neural population adapt to different sensory contexts with expected shifts in $p(z)$. Using our NSC framework, we would expect to be able to predict how these neuronal responses should change (as reflected in updated $r_{new} \sim p_{new}(z|x)$) under new contexts.

---

### Decision · Program_Chairs · 2023-09-21

**Decision:**

Accept (poster)

**Comment:**

This paper describes a new data-driven approach for quantitatively assessing the neural sampling hypothesis. This question is timely and important, but the paper elicited a major difference in opinion from the four reviewers, with three giving a "weak accept" and one giving a "strong reject" score. After reading the reviews, author rebuttal, and discussion, I am inclined to side with the three reviewers who voted for acceptance. Reviewer GaXp raises a worthwhile point in criticizing the lack of biologically plausible implementations of neural sampling codes, which I feel the authors should certainly mention in their discussion as a shortcoming of the theory and an important avenue for future work. However, I ultimately agree with the authors that the lack of such an implementation is not a fatal flaw, and that a framework for quantitatively assessing the (normative) sampling hypothesis based codes is a major contribution in its own right. (Indeed, if we go back to the original sparse coding papers from Olshausen & Field, the normative theory made an important contribution despite the lack of plausible implementation; in my reading, it was only later that papers such as Rozzell et al 2008 took the additional steps toward proposing biologically realistic implementations of sparse coding). So ultimately I would like to recommend this paper for acceptance. However, all reviewers raised concerns about clarity (e.g., about the relationship between latents z and r), and so I would urge the authors to revise the paper to address all reviewer comments carefully and thoroughly.